# Black phosphorus ink formulation for inkjet printing of optoelectronics and photonics

Guohua Hu [1], Tom Albrow-Owen[1], Xinxin Jin[2], Ayaz Ali[3], Yuwei Hu[2], Richard C.T. Howe [1], Khurram Shehzad [3], Zongyin Yang [1], Xuekun Zhu[2], Robert I. Woodward [4], Tien-Chun Wu[1], Henri Jussila[5], Jiang-Bin Wu[6], Peng Peng[7,8], Ping-Heng Tan [6], Zhipei Sun[5], Edmund J.R. Kelleher [4], Meng Zhang[2,8], Yang Xu[3] & Tawfique Hasan[1]

Black phosphorus is a two-dimensional material of great interest, in part because of its high carrier mobility and thickness dependent direct bandgap. However, its instability under ambient conditions limits material deposition options for device fabrication. Here we show a black phosphorus ink that can be reliably inkjet printed, enabling scalable development of optoelectronic and photonic devices. Our binder-free ink suppresses coffee ring formation through induced recirculating Marangoni flow, and supports excellent consistency (< 2% variation) and spatial uniformity (< 3.4% variation), without substrate pre-treatment. Due to rapid ink drying (< 10 s at < 60 °C), printing causes minimal oxidation. Following encapsulation, the printed black phosphorus is stable against long-term (> 30 days) oxidation. We demonstrate printed black phosphorus as a passive switch for ultrafast lasers, stable against intense irradiation, and as a visible to near-infrared photodetector with high responsivities. Our work highlights the promise of this material as a functional ink platform for printed devices.

---

[1] Cambridge Graphene Centre, University of Cambridge, Cambridge CB3 0FA, UK. [2] School of Electronic and Information Engineering, Beihang University, Beijing 100191, China. [3] School of Information Science and Electronic Engineering, Zhejiang University, Hangzhou 310027, China. [4] Femtosecond Optics Group, Department of Physics, Imperial College London, London SW7 2AZ, UK. [5] Department of Electronics and Nanoengineering, Aalto University, Tietotie 3, Espoo FI-02150, Finland. [6] State Key Laboratory of Superlattices and Microstructures, Institute of Semiconductors, Chinese Academy of Sciences, Beijing 100083, China. [7] School of Mechanical Engineering and Automation, Beihang University, Beijing 100191, China. [8] International Research Institute for Multidisciplinary Science, Beihang University, Beijing 100191, China. Correspondence and requests for materials should be addressed to M.Z. (email: mengzhang10@buaa.edu.cn) or to Y.X. (email: yangxu-isee@zju.edu.cn) or to T.H. (email: th270@cam.ac.uk)

Black phosphorus (BP) is a two-dimensional (2d) material with unique optoelectronic properties[1, 2]. These include high carrier mobility (up to 50,000 cm$^2$ V$^{-1}$ s$^{-1}$ in bulk at 30 K[2]) and a thickness-dependent direct bandgap, transitioning from ~0.3 eV in bulk to ~2 eV in mono-layer[2]. These properties suggest potential applications in optoelectronics and photonics, in particular for the development of devices such as transistors, light emitting diodes, photodetectors, solar cells and all-optical switches for ultrafast lasers[1–6]. Thus far, mechanical exfoliation has been the dominant production method for BP used in fundamental studies and small-scale device demonstrations. However, the scope of this technique is limited due to low yield and a lack of control[1, 3, 7]. For practical applications, a potential route is ultrasound-assisted liquid phase exfoliation (UALPE) of bulk BP crystals. This supports the production of dispersions enriched with thin (i.e. mono- and few-layer) flakes that can be exploited for their optoelectronic properties[1, 3, 7, 8]. Provided stable dispersions can be realised, existing printing processes can be adopted for large-scale, high speed device fabrication[9]. Inkjet printing is one such printing process that enables high resolution (~50 μm) maskless patterning[10]. Indeed, inkjet printing has been used with other 2d materials such as graphene and molybdenum disulphide (MoS$_2$) to demonstrate novel devices, including transistors[11–16], photodetectors[14, 15, 17] and photovoltaics[14, 18].

To date, however, there are no reports of inkjet printing of BP. A key reason lies in the solvents suitable for UALPE production of BP. These solvents typically have a high boiling point, for instance N-methyl-2-pyrrolidone (NMP)-204 °C and N-cyclohexyl-2-pyrrolidone (CHP)-284 °C[1, 3, 8]. A high boiling point can lead to long drying times when depositing the BP dispersions, resulting in significant oxidation and hence preventing the fabrication of stable devices under ambient conditions[1, 3, 19]. In addition, inkjet printing relies on three steps: stable jetting of single-droplets[10, 20], appropriate wetting of the substrate[21, 22] and uniform material distribution during droplet drying[23, 24]-all of which pose challenges for BP dispersions in these solvents. While favourable for exfoliation, these solvents have neither suitable surface tension nor viscosity for inkjet printing[11, 17], which may result in unstable jetting. Also, these solvents do not possess suitable surface tension to wet commonly used substrates such as Si/SiO$_2$, glass and polyethylene terephthalate (PET) after deposition, resulting in non-uniform or even discontinuous material deposition[18, 25]. Finally, driven by solvent evaporative losses at the droplet edges during drying, an outward fluid flow may form within the droplet, leading to the so-called 'coffee ring' effect, where flakes concentrate at the droplet edges[23, 24], compromising the printing uniformity[15]. The above issues can be alleviated by including additives such as polymer binders into the UALPE dispersions, as demonstrated for other 2d materials (e.g. graphene[12, 26, 27] and MoS$_2$[13]). However, unlike solvents, binders form an integral part of the printed film, and must be removed through high temperature annealing[12, 13, 26] or intense pulsed light[27] to retain the functionalities of the 2d materials for optoelectronic and photonic devices. This approach is impractical here since these processes will likely lead to BP oxidation while exposed to ambient conditions. We also note that in typical UALPE dispersions, the relatively low concentration of BP (< 1 gL$^{-1}$ [1, 3, 8]) requires repeated printing to deposit sufficient material, lengthening device fabrication times.

Here, we demonstrate a binder-free inkjet ink composed of UALPE produced BP and a binary solvent carrier of isopropyl alcohol (IPA) and 2-butanol. The ink formulation allows stable jetting, induces recirculating Marangoni flow to control the coffee ring effect, and ensures wetting of untreated substrates (Si/SiO$_2$, glass and PET). The low boiling point of the alcohols promotes a rapid ink drying (< 10 s at < 60 °C). By raising the ink concentration, we reduce the number of printing repetitions required to deposit sufficient BP for device fabrication. The rapid ink drying and reduced printing repetitions lead to a reduced window of time available for the oxidation of BP under ambient conditions during printing. Through optimisation of the printing characteristics, our formulated ink allows high printing consistency (< 2% variation across printing repetitions) and spatial uniformity (< 3.4% variation across printed patterns). After encapsulation with parylene-C, the printed BP remains stable for > 30 days under ambient conditions. Combining the printing consistency and uniformity, this long-term stability allows us to develop robust photonic and optoelectronic devices using printed BP, including a saturable absorber (SA) for stable

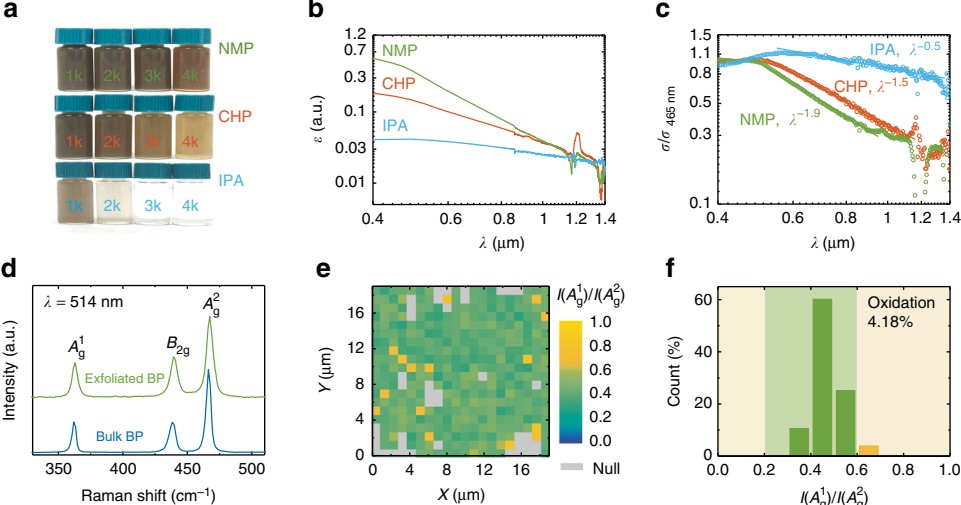

**Fig. 1** BP exfoliation and characterisation. **a** Photographs of the BP dispersions in NMP, CHP and IPA, centrifuged at 1–4 krpm. **b** Optical extinction (log–log scale) of the BP dispersions (centrifuged at 4 krpm), the dispersions are diluted to 10 vol% to avoid detector saturation. **c** Optical scattering with associated fitting (log–log scale) of the dispersions. The scattering is normalised to the 465 nm extinction peak. **d** Raman spectrum for exfoliated and bulk BP, with intensity normalised to that of the $A_g^2$ peak, $I\left(A_g^2\right)$. **e** Raman map of the intensity ratio, $I\left(A_g^1\right)/I\left(A_g^2\right)$, with 1 μm spatial step. The *grey squares* correspond to regions where the Raman intensity is too low for accurate interpretation. **f** Corresponding histogram of the map in **e**

generation of ultrashort pulses under an intense irradiation of $32.7\,MW\,cm^{-2}$ for $> 30$ days, and a broadband photodetector device for visible and near-infrared wavelengths with a responsivity of up to $164\,mA\,W^{-1}$. These demonstrations highlight the suitability of our process for the fabrication of stable BP-based devices, confirming its potential as a platform for the development of future optoelectronic and photonic technologies.

## Results

**Production of BP dispersions.** We produce dispersions of thin BP flakes via UALPE, as detailed in Methods. The choice of solvent is key to achieving effective exfoliation and stable dispersions, since UALPE is reliant on optimising the intermolecular interactions with the BP flakes to minimise the enthalpy of mixing[28–30]. Recent investigations on UALPE of BP show that the suitable solvents that possess a matched surface tension ($\sim 40\,mN\,m^{-1}$) to facilitate exfoliation without reaggregation are typically high boiling point organic solvents, such as NMP and CHP[1, 3, 8]. We therefore produce the BP dispersion in both these solvents as a starting point for our ink formulation. We additionally investigate lower boiling point solvents, for instance IPA (82.6 °C), which has previously been shown to produce meta-stable dispersions of graphene, despite a mismatch in surface tension[30]. Figure 1a shows photographs of the as-prepared BP dispersions centrifuged at 1–4 krpm. At a higher centrifugation speed, the increased sedimentation force leads to lower concentration of dispersed flakes, and hence a more colourless dispersion. The distributions of flake size (thickness and lateral dimension) in the dispersion vary with the centrifugation speed, as less exfoliated larger and thicker flakes sediment more readily[31–33]. We therefore use the dispersions centrifuged at 4 krpm for the remainder of this work. The optical extinction spectrum (log-log scale) for each dispersion (Fig. 1b) has a peak at ~465 nm, with an approximately linear decrease at longer wavelengths ($> 500$ nm). Using the extinction coefficient, $267\,L\,g^{-1}\,m^{-1}$ at 660 nm[3], we estimate the concentrations of the NMP, CHP and IPA dispersions as 0.54, 0.32 and $0.13\,gL^{-1}$, respectively. We attribute the extinction variations observed at the near-infrared region to the ambient moisture absorbed by the solvents during the exfoliation process (Supplementary Fig. 1a).

Preceding studies on other 2d materials show the extinction has a scattering component proportional to $\lambda^{-n}$, where $\lambda$ is the wavelength and $n$ is the scattering exponent[31, 33]. We evaluate this scattering by subtracting the absorbance obtained using an integrating sphere from the measured extinction. The scattering (Fig. 1c) at the longer wavelength region is fitted to $\lambda^{-1.9}$ (NMP), $\lambda^{-1.5}$ (CHP) and $\lambda^{-0.5}$ (IPA). We note that $n < 4$ corresponds to Mie scattering[34, 35], allowing estimation of the characteristic dimensional length of the dispersed flakes as ~80–210 nm (Supplementary Note 1), typical for UALPE flakes of 2d materials[28, 29, 36–40]. Meanwhile, the larger $n$ of the NMP dispersion indicates that BP is better exfoliated (producing thinner and smaller flakes) in NMP than in either CHP or IPA[31, 33, 34]. These are confirmed by the flake size distributions characterised by atomic-force microscopy (AFM) (Supplementary Fig. 2). We therefore use the NMP dispersion for ink formulation. The average flake thickness in NMP is 3.37 nm, ~6 layers (considering 0.9 nm for the first single layer and 0.5 nm for subsequent individual layers[19, 41]).

We next characterise the proportion of oxidised BP flakes in the NMP dispersion via Raman spectroscopy. The Raman spectrum of BP typically contains three major peaks close to $400\,cm^{-1}$, corresponding to one out-of-plane $\left(A_g^1\right)$ and two in-plane ($B_{2g}$, $A_g^2$) vibrational modes[19, 42]. Previous studies show that BP exhibits highly anisotropic electron-phonon interactions, making the Raman peaks polarisation-, wavelength- and thickness-dependent[8, 43]. However, we show that this polarisation behaviour can be nullified when the studied BP flakes (dropcast randomly distributed BP flakes in this case) are not aligned in orientation (Supplementary Fig. 3a). Figure 1d shows a typical Raman spectrum for our bulk BP and exfoliated flakes (NMP dispersion dropcast onto Si/SiO$_2$ and dried). The exfoliated sample shows $A_g^1$ at $\sim 363.0\,cm^{-1}$, $B_{2g}$ at $\sim 439.3\,cm^{-1}$ and $A_g^2$ at $\sim 467.4\,cm^{-1}$, consistent with previous studies on mechanical[19, 42] and solution[1, 3, 7, 8, 44] exfoliated BP. Since the peaks are due to BP crystalline lattice vibrations[42], this consistency suggests that our exfoliated BP flakes are highly crystalline. Statistical investigation (~360 measurements) on the full width at half maximum of these three peaks also notes no discrepancy from those reported in literature[19] (Supplementary Fig. 4), further suggesting high crystallinity. These three peaks show a blue-shift compared to the bulk BP (~361.9, ~438.3 and ~466.4 cm$^{-1}$, respectively), also consistent with previous studies[19, 42, 44]. However, the observed blue-shift (0.8–1.3 cm$^{-1}$) is smaller than that of atomically thin ($< 4$ layers) samples[44]. This indicates that the layer number of our exfoliated BP is larger than 4, in agreement with the AFM results. The intensity ratio of $A_g^1$ and $A_g^2$, $I\left(A_g^1\right)/I\left(A_g^2\right)$ which we demonstrate to be polarisation insensitive (Supplementary Fig. 3b), has been used as an indication for the oxidation levels of exfoliated BP flakes, with a range 0.2–0.6 for minimal oxidation (Supplementary Table 1)[1, 19]. Figure 1e shows the Raman map of $I\left(A_g^1\right)/I\left(A_g^2\right)$ for a typical region of exfoliated BP flakes deposited onto Si/SiO$_2$, showing no localised clusters with $I\left(A_g^1\right)/I\left(A_g^2\right) > 0.6$ (see the measurement scheme in Supplementary Fig. 5 and associated discussion in Supplementary Note 2). Figure 1f shows the corresponding histogram, showing only 4.2% of the measured data points fall outside the 0.2–0.6 range (*yellow background*). This low oxidation proportion demonstrates that our UALPE process causes minimal oxidation. We note that this is significantly lower than previous reports on solution exfoliated BP, for instance in Hanlon et al.[1], where all of the values fall outside of 0.2–0.6.

**Formulation of BP ink.** We now consider inkjet printing of BP, where an ink must be designed for stable jetting and appropriate substrate wetting, and be able to dry to produce a spatially uniform material deposition. Stable jetting in this case is defined as a single droplet jetted for each electrical impulse with no secondary droplet formation[10]. Unstable jetting is undesirable, as it can lead to material deposition onto untargeted areas. Stable jetting is dependent on the ink viscosity ($\eta$, mPa s), surface tension ($\gamma$, mN m$^{-1}$) and density ($\rho$, g cm$^{-3}$) as well as cartridge nozzle diameter ($D$, μm), typically combined into an inverse Ohnesorge number, $Z = \sqrt{\gamma\rho D}/\eta$[10, 11, 20]. A $Z$ value of 1–14 indicates stable jetting, with $Z < 1$ indicating an ink that will not jet, and $Z > 14$ an ink prone to generating secondary droplets[10, 11, 20]. Solvents suitable for UALPE such as NMP tend to give $Z > 14$[11, 17] due to their low $\eta$ (~2 mPa s) and high $\gamma$ (~40 mN m$^{-1}$). While jetting is possible under these conditions through control of the electrical impulses[11, 17], it is preferable to formulate the BP ink with an optimal $Z$ value for consistent printing and high device yield.

A rule of thumb for printing is that $\gamma$ of the ink should be 7–10 mN m$^{-1}$ lower than the substrate surface energy[22], otherwise the deposited droplets will not wet the substrate to form a continuous coating[21, 22]. Solvents such as NMP ($\gamma \sim 40$ mN m$^{-1}$) and CHP ($\gamma \sim 43$ mN m$^{-1}$) are therefore not suitable for direct printing on commonly used substrates such as Si/SiO$_2$

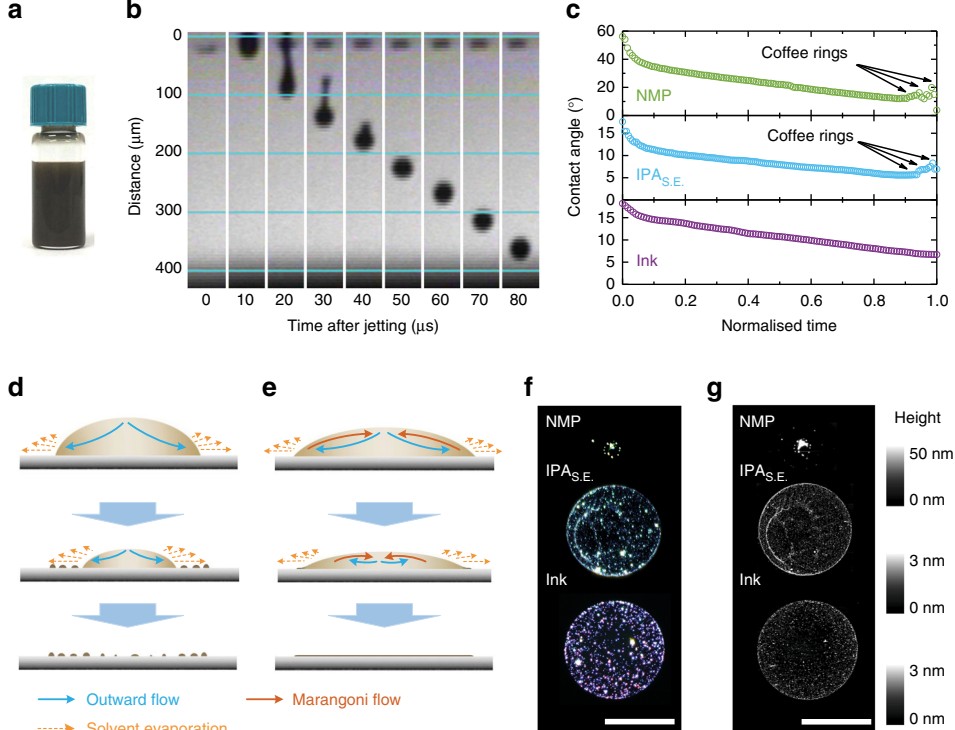

**Fig. 2** BP ink jetting and drying. **a** Photograph of formulated BP ink. **b** Droplet jetting sequence observed from the printer stroboscopic camera. **c** Change in contact angle for the NMP dispersion, the BP-IPA$_{S.E.}$ (after solvent exchange) and the BP ink on Si/SiO$_2$ during the drying process. The horizontal axis is normalised to the drying time of each droplet; droplet drying process **d** without and **e** with a recirculating Marangoni flow induced to prevent coffee ring effect. **f** Optical micrographs and **g** AFM images of the dried droplets, *scale bar*, 50 μm. The contrast in **f** has been enhanced for clarity

and glass (SiO$_2$ ~36 mN m$^{-1}$[45]) and PET (~48 mN m$^{-1}$[46]). Indeed, NMP-based 2d material inks were shown to render non-uniform and discontinuous printed patterns on untreated Si/SiO$_2$[11, 14]. The BP ink should therefore be formulated with $\gamma < 30$ mN m$^{-1}$ to allow wetting of the substrates.

After printing, the droplet drying process is vital for uniform material deposition. As discussed, the coffee ring effect arises from an unbalanced flow within the droplet during the drying process[23, 24]. During the drying process for typical UALPE droplets with a single solvent, the higher surface area to volume ratio at the droplet edges causes more rapid solvent evaporation than in the droplet centre. This leads to an outward flow from the centre to the edges to replenish the evaporated solvents that carries the dispersed flakes with it[23, 24]. With no recirculating flow, the flakes remain at the edges, not only forming a ring, but also preventing the droplet from receding further, leaving little to no material at the centre of the dried droplet. One way that this can be overcome is to induce a secondary recirculating flow to balance this outward flow. Previous studies show that creating a composition variation across the droplet may give rise to a surface tension driven (Marangoni) recirculating flow[47–51]. For example, in an ink composed of a binary solvent mixture, the differing evaporation rates of the two solvents lead to variations in the solvent proportions across the droplet, where the proportion of the faster evaporating solvent is highest at the centre and the slower evaporating solvent at the edges[47, 51]. This may result in a temperature gradient in the droplet due to latent heat of vapourisation and hence a recirculating surface tension gradient, inducing a recirculating Marangoni flow[48–50]. By recirculating the dispersed flakes, this flow prevents coffee ring formation, ensuring a uniform material deposition.

Previous studies show water- and alcohol-based dispersions of other 2d materials are suitable for inkjet printing[12, 13, 15, 18, 25].

For instance, while our work was in review, a biocompatible water-based ink formulation of 2d materials for large area printing was reported by McManus et al.[15]. However, developing stable aqueous BP dispersions requires deoxygenated water assisted by surfactants[7], requiring additional removal steps after printing. Note that given their relatively low hazardous potential, alcohols such as IPA are widely used in various commercial graphics and functional inks[52]. The low surface tensions of alcohols favour substrate wetting, and their low boiling points allow fast ink drying, critical for inkjet printing of BP. To formulate the BP ink, we therefore transfer the BP flakes into anhydrous IPA through a solvent exchange process (see Methods). To distinguish from the previous IPA dispersion, we refer to BP redispersed in IPA as BP-IPA$_{S.E.}$. The solvent exchange process concentrates BP-IPA$_{S.E.}$, about 10 times that of the NMP dispersion. A secondary alcohol, 2-butanol (boiling point 100 °C), is added at 10 vol% to formulate the BP ink (Fig. 2a). The 10 vol% secondary alcohol is included not only to induce a recirculating Marangoni flow, but also to preserve a high printing resolution (see Supplementary Fig. 6 and associated discussion in Supplementary Note 3). The ink has a concentration ~5 gL$^{-1}$, verified via optical extinction (Supplementary Fig. 7a). The optical scattering of the ink (Supplementary Fig. 7b) is not significantly different from the NMP dispersion, indicating that the ink formulation steps do not induce aggregation of the BP flakes. The ink shows a high stability, with 1% flake sedimentation during 1 week, over a timeframe that is viable for large-scale ink production and printing (Supplementary Fig. 8a). The ink has $\eta \sim 2.2$ mPa s, $\gamma \sim 28$ mN m$^{-1}$, $\rho \sim 0.8$ g cm$^{-3}$, giving $Z \sim 10$ ($D = 22$ μm), well within the optimal $Z$ value range for stable jetting. This is confirmed by the jetting sequence without the formation of secondary droplets (Fig. 2b). The low ink surface

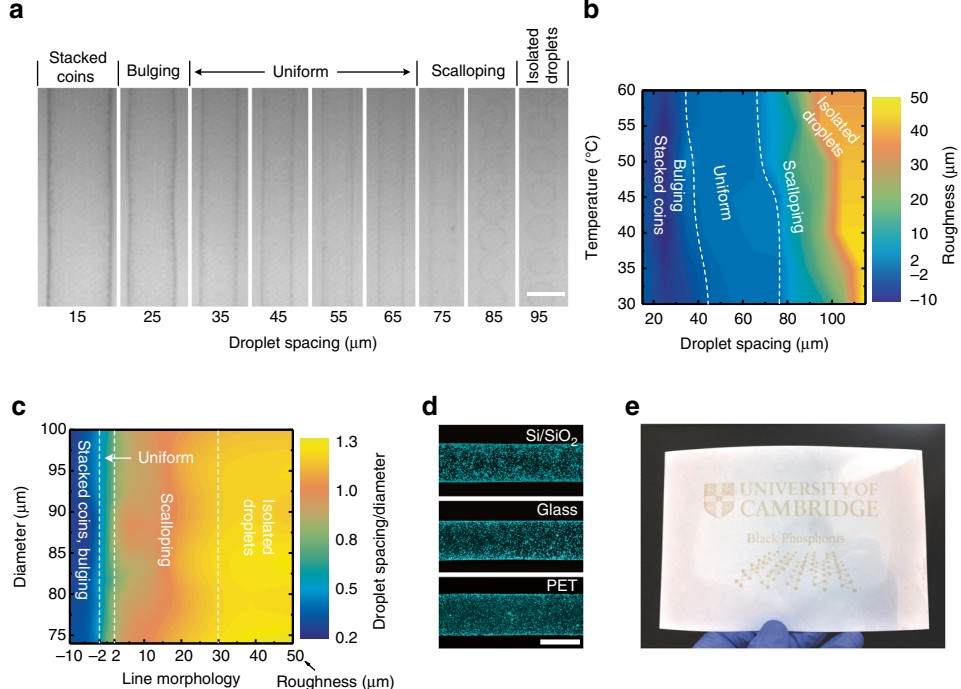

**Fig. 3** Optimisation of BP printing conditions. **a** BP printed on Si/SiO₂ at 60 °C showing the effect of droplet spacing on line morphology, photos taken by printer fiducial camera, *scale bar*, 100 μm. **b** Effect of droplet spacing and printing temperature on the roughness along line edges, the roughness from uniform to stacked coins is defined as negative. **c** Ratio of droplet spacing to dried droplet diameter in each line morphology regions under varied diameters. **d** Dark field optical micrographs of optimised printed tracks on Si/SiO₂, glass and PET, *scale bar*, 100 μm. The contrast has been enhanced for clarity. **e** Inkjet printed BP on untreated ultrathin PET (1.5 μm) over an area of 100 mm × 63 mm. The ultrathin PET is laminated onto photopaper for the ease of handling

tension gives a contact angle < 30° on our chosen substrates (Si/SiO₂, glass, PET), indicating good wetting.

To understand the drying process, we study the deposited droplets using time-dependent contact angle measurements (Fig. 2c). As discussed above, when deposited onto substrates, fluid initially flows outwards in all the three droplets due to higher solvent evaporation rates at the edges. Accordingly, the contact angle in these three cases show descending trends against the drying time, without noticeable variations. However, as depicted in Fig. 2d, the lack of recirculating Marangoni flow in NMP and IPA$_{S.E.}$ causes BP flakes to concentrate at the contact edges. This prevents the droplet from receding further until the contact angle decreases beyond a lower threshold, where the droplet rapidly contracts and leaves a series of material rings (Fig. 2f, g), observed as sharp variations in the contact angle. No such variation is seen for the case of our ink, indicating the presence of a recirculating Marangoni flow as depicted in Fig. 2e to prevent coffee ring formation. Indeed, the ink forms an even BP flake distribution devoid of any visible coffee rings in the dark field micrographs and the AFM images (Fig. 2f, g).

**Optimisation of printing conditions**. We next determine the optimal printing conditions (such as the substrate temperature and droplet spacing), which can define the printing morphology[53]. Figure 3a shows the morphology of printed lines on untreated Si/SiO₂ at 60 °C with varied droplet spacing. When the droplet spacing is 15 μm, the line is broad as the droplets significantly overlap (termed as stacked coins), causing the ink to spread further across the substrate. As the droplet spacing increases, the line morphology first becomes bulging (due to excess ink in the droplet merging), before forming a line with uniform edges (droplet spacing 35 μm). Further increase in

the droplet spacing leads to a narrower scalloped line (droplet spacing 75 μm) due to insufficient ink to merge, and ultimately (droplet spacing 95 μm) to a series of isolated droplets since they are too far apart to merge. From the images in Fig. 3a we can determine the line-edge roughness under varied droplet spacing (see the estimation scheme in Supplementary Fig. 9a, b and associated discussion in Supplementary Note 4). The roughness of stacked coins and bulging is defined as negative to differentiate from that of scalloping and individual droplets. A roughness < ±2 μm is defined as uniform.

The substrate temperature influences the dried droplet diameter (Supplementary Fig. 9c), which defines the droplet merging behaviour and hence the printing morphology. We therefore repeat this study for different substrate temperature allowing us to generate a map of the edge roughness under varied printing conditions (Fig. 3b). As indicated by the shape and orientation of the highlighted uniform region in the map, the droplet spacing is the dominant factor in defining the morphology. Essentially, the formation of stacked coins, bulging, uniform, scalloped lines and isolated droplets is dependent on the ratio of the droplet spacing to the diameter. We present this ratio value at the different morphology regions with respect to the varied diameters, showing that the uniform region corresponds to a ratio value ~0.5–0.8 (Fig. 3c). This provides a general guidance for inkjet printing of our BP ink: under different printing conditions, the droplet spacing should be 0.5–0.8 of the dried droplet diameter.

According to this criterion, we choose a droplet spacing of 35 μm and a substrate temperature of 60 °C since these printing conditions not only allow optimal morphology but also rapid ink drying (< 10 s observed from the printer fiducial camera), critical for avoiding BP degradation. Using dark-field optical microscopy, we examine the BP flake distributions in lines printed onto

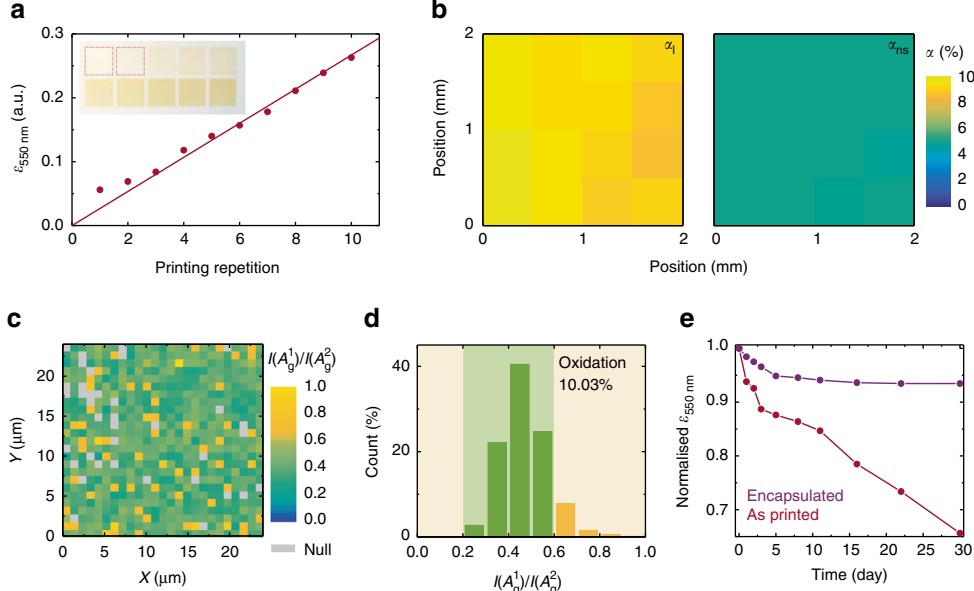

Fig. 4 Characterisation of printed BP. **a** Optical extinction (at 550 nm for printed BP with 1–10 printing repetitions, inset—photograph of printed BP squares (8 mm × 8 mm). **b** Spatial linear ($\alpha_l$) and nonsaturable ($\alpha_{ns}$) optical absorption (at 1562 nm) across printed BP, spatial step 0.5 mm. **c** Raman map of the intensity ratio, $I\left(A_g^1\right)/I\left(A_g^2\right)$, for BP printed onto Si/SiO$_2$ with 1 μm spatial step. The *grey squares* correspond to regions where the Raman intensity is too low for accurate interpretation; **d** Corresponding histogram of the map in **c**. **e** Change in optical extinction (at 550 nm) for encapsulated and unencapsulated printed BP across 30 days under ambient conditions

Si/SiO$_2$, glass and PET (Fig. 3d). As shown, the BP flakes are evenly deposited along the length and the width of the lines, without any noticeable coffee rings. This demonstrates that our ink is suitable for inkjet printing onto commonly used substrates without requiring any surface treatment. Moving on from individual lines, we investigate printing of BP on larger scales. Figure 3e shows a uniform, high resolution inkjet printed BP pattern on untreated ultrathin PET (1.5 μm) over a 100 mm × 63 mm area. This demonstrates that our BP ink can potentially fabricate printable devices on a large scale without the loss of printing resolution and material uniformity. We also note that the printed BP is free of any additives (e.g. polymer binders), thus post-printing treatments that may oxidise the BP flakes (e.g. high temperature annealing) are avoided.

**Characterisation of inkjet printed BP**. We next study the printing stability of our BP ink. We first assess the printing consistency with respect to the number of repetitions. Figure 4a shows the optical extinction of printed BP squares on untreated glass with 1–10 printing repetitions. The extinction increases linearly with the printing repetition, with < 2% variation, demonstrating an excellent printing consistency and an ability to control the optical density of the deposited BP through printing repetitions. We attribute the deviation of extinction for one to two printing repetitions to a modification in the air-glass interface after printing. For successive printing repetitions the trend follows a constant increase with each printing repetition, showing high reproducibility (Supplementary Fig. 9d). We then assess the spatial uniformity of an individual printed BP square by mapping optical absorptions using raster-scanning (Supplementary Fig. 10; Supplementary Note 5). This allows us to quantify its linear absorption ($\alpha_l$) under low irradiation intensities, and nonsaturable absorption ($\alpha_{ns}$) under high intensities (Fig. 4b). The average $\alpha_l$ and $\alpha_{ns}$ is $9.19 \pm 0.31\%$ and $4.99 \pm 0.09\%$, respectively, with a variation of < 3.4%. This demonstrates a high spatial uniformity. The consistency between printing repetitions and the uniformity across printed patterns, as well as the large-scale printability

confirm that our BP ink allows for stable, repeatable and reliable inkjet printing.

The stability of printed BP against oxidation is key to ensuring stable operation of BP-based devices. Having shown that our exfoliated BP flakes have a low oxidation proportion (4.2%), here we assess the oxidation of BP following inkjet printing. Figure 4c shows the Raman map of $I\left(A_g^1\right)/I\left(A_g^2\right)$ for a typical printed BP region on Si/SiO$_2$. The corresponding histogram is in Fig. 4d. As shown, the printed BP still has a low oxidation (10%), indicating that inkjet printing of BP does not adversely affect the BP stability. We attribute this small increase to oxidation during the solvent exchange and inkjet printing steps. We also note that printed BP shows no localised clusters where $I\left(A_g^1\right)/I\left(A_g^2\right) > 0.6$.

Having confirmed that the deposition process does not induce significant oxidation, it is necessary to prevent further degradation of the printed BP. Typical methods include encapsulating BP with polymers such as polydimethylsiloxane[54] and parylene-C[19]. However, these methods have only been shown as effective for mechanically exfoliated individual BP flakes, which are atomically smooth. Successful encapsulation requires a pin-hole free layer, thus printed BP presents a greater challenge, due to the relatively small flake size and high roughness, characteristic of deposited films. Indeed, to the best of our knowledge, successful encapsulation for long-term stability has never been reported for solution deposited BP samples. However, we expect that parylene-C—widely used in the microelectronics industry, as a chemically inert, pin-hole free and uniform passivation layer[55, 56]—is likely to be effective since it forms a conformal coating even on rough surfaces. We therefore investigate encapsulation with parylene-C to preserve our printed BP against oxidation. Figure 4e shows the optical extinction at 550 nm of printed BP samples over 30 days when exposed to ambient conditions; see Supplementary Fig. 11 for the extinction change at 350–850 nm. The extinction decreases rapidly for the unencapsulated sample due to BP oxidation over time and continues its downward trend at the end of this

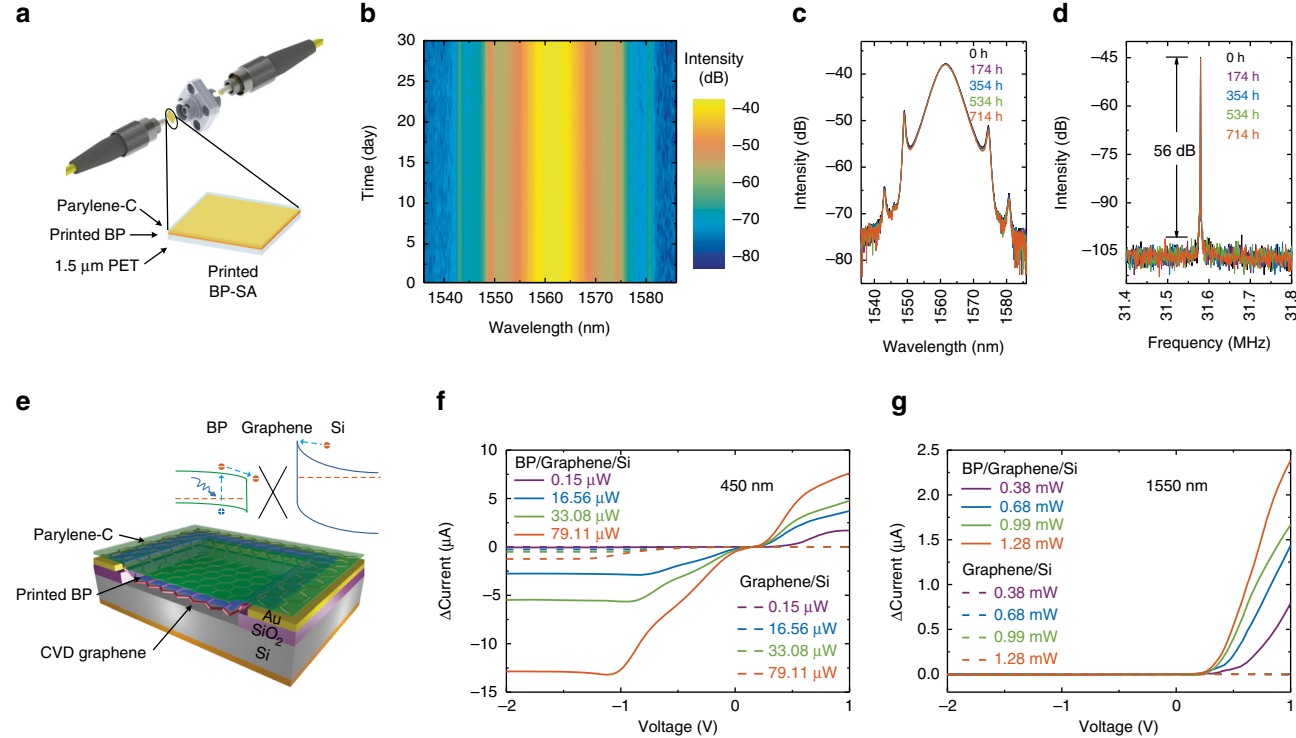

**Fig. 5** Optoelectronic and photonic devices using inkjet printed BP. **a** Integration of a printed BP saturable absorber (SA) between fibre patch cords for ultrashort pulse generation with fibre lasers. **b** Output laser spectrum across 30 days, and **c** overlay of the spectrum acquired after 0, 174, 354, 534 and 714 h of operation. **d** Overlay of radio frequency spectrum at the cavity fundamental repetition frequency of 31.6 MHz after 0, 174, 354, 534 and 714 h of operation. **e** Schematic of the BP/graphene/Si Schottky junction photodetector and the device band diagram configuration. Associated current response for **f** 450 nm and **g** 1550 nm illumination, the *dashed lines* show the response of the graphene/Si Schottky junction photodetector without printed BP. Dark current has been subtracted in all the cases

measurement period. For the encapsulated sample (100 nm thick parylene-C), the extinction shows a slow decrease (5%) during the first 5 days. However, the sample is then stabilised and shows no noticeable changes for the remaining measurement period. We suggest this initial decrease is due to trapped moisture and oxygen in the printed BP sample. We therefore conclude that the encapsulated inkjet printed BP has long-term stability.

We further study the stability of the encapsulated sample through $I\left(A_g^1\right)/I\left(A_g^2\right)$. The proportion of $I\left(A_g^1\right)/I\left(A_g^2\right)$ outside of 0.2–0.6 increases from 10% to 23% once encapsulated, but then stabilises at ~33% (Supplementary Fig. 12). We note that the oxidation threshold of $I\left(A_g^1\right)/I\left(A_g^2\right)$ for BP encapsulated with parylene-C is yet to be reported. To investigate this further, we measure $I\left(A_g^1\right)/I\left(A_g^2\right)$ of freshly cleaved bulk BP crystal with and without encapsulation. We find that immediately after encapsulation the proportion of $I\left(A_g^1\right)/I\left(A_g^2\right)$ outside of 0.2–0.6 increases from 3.1% to 5.8% (Supplementary Fig. 13). However, this increase is unlikely to be a result of oxidation of the bulk BP crystal. We suggest that this is due to interactions between BP and parylene-C; also see the discussion in Supplementary Note 6. Favron et al.[19] suggest that $I\left(A_g^1\right)$ may be sensitive to perturbations from contacting substances, while $I\left(A_g^2\right)$ is not. We therefore propose that this sizeable increase in $I\left(A_g^1\right)/I\left(A_g^2\right)$ in our printed sample after encapsulation is partially as a result of the increase in $I\left(A_g^1\right)$. This means that the absolute values of

$I\left(A_g^1\right)/I\left(A_g^2\right)$ may not correctly represent the oxidation proportion in BP samples coated with parylene-C. Additionally, the larger increase in $I\left(A_g^1\right)/I\left(A_g^2\right)$ values for printed BP may arise from the greater surface area of the exfoliated flakes in contact with parylene-C. We therefore conclude that the encapsulated printed BP is well protected from degradation and can be used to fabricate and operate devices under ambient conditions.

**Printed BP-based optoelectronic and photonic devices**. We now investigate the performance of the inkjet printed BP when integrated into optoelectronic and photonic devices. The strong saturable optical absorption (Fig. 4b, Supplementary Fig. 10) suggests the potential for printed BP in photonics, e.g. as SAs to mode-lock ultrafast lasers[40, 57, 58]. A well-developed method to fabricate 2d material-based SAs is to produce polymer composites through blending dispersions with polymers and subsequently evaporating the solvents[37–40, 59]. However, the evaporation process can be time consuming, leading to BP oxidation. In addition, the reported BP-based SAs have thus far shown very limited usable lifetime (< 28 h; Supplementary Table 2), likely due to oxidation under intense laser irradiation. Inkjet printing of BP directly onto optical components such as quartz, polymers and optical fibres may provide an alternative, allowing rapid device fabrication, controllable optical absorption and spatial uniformity. To fabricate the SA, BP is inkjet printed onto the ultrathin PET and subsequently encapsulated with 100 nm thick parylene-C. The BP-SA is then introduced into the cavity of an erbium-doped fibre laser for the generation of ultrashort pulses through mode-locking. See the measurement detail in Supplementary Note 7. The output laser pulse spectrum across 30 days

(Fig. 5b) and the overlay of the laser spectrum acquired after 0, 174, 354, 534 and 714 h of operation (Fig. 5c) show no variations, demonstrating an excellent operation stability. The radio frequency (RF) spectrum at the fundamental frequency also shows a high stability across the 30 days (Fig. 5d, Supplementary Fig. 14b). The high intensity compared to background (~53 dB) indicates an excellent mode-locking stability[60]. With regards to BP stability under intense irradiation, our measurement condition is over three orders of magnitude higher than that reported by Favron et al.[19]. The generation of stable ultrashort pulses, over a much longer operation period (> 24 times longer), demonstrates that inkjet printing of BP can enable scalable fabrication of high performance and long-term stable photonic devices.

As discussed, BP is an ideal material for visible and near-infrared optoelectronics, including photodetectors[61–64]. Recently, graphene/Si Schottky junction photodetectors (Gr/Si) have attracted significant interest due to their advantages-ambient condition operation and high responsivity[65–68]. However, the operating wavelength range is limited by the bandgap of Si (1.1 eV). Combining this device structure with functional materials, it is possible to tune the Gr/Si Schottky junction barrier and enhance optical absorption for improved detection performance[65]. We propose that BP is ideally suited to this role. We combine our inkjet printed BP with a Gr/Si device (Supplementary Fig. 15) to fabricate a BP/graphene/Si Schottky junction photodetector (BP/Gr/Si) (Fig. 5e). See the measurement detail in Supplementary Note 8. Figure 5f, g shows the photocurrent (after dark current subtraction) of BP/Gr/Si and Gr/Si under 450 and 1550 nm excitation, respectively. As shown, Gr/Si shows a reverse responsivity ~16 mA W$^{-1}$ and no forward response at 450 nm. However, it does not respond to 1550 nm excitation, as expected since 1550 nm (0.8 eV) falls below the bandgap of Si. However, BP/Gr/Si not only produces a significantly larger reverse responsivity (~164 mA W$^{-1}$) but also a forward responsivity (~95 mA W$^{-1}$) under 450 nm excitation. Meanwhile, the device exhibits a forward responsivity ~1.8 mA W$^{-1}$ under 1550 nm excitation, demonstrating that printed BP enables the Gr/Si device to work in the near-infrared region. The performance enhancement indicates that BP improves optical absorption of the hybrid structure. In addition to the thermal effect for Schottky based devices[68] that possibly exists in BP/Gr/Si as a result of the increased optical absorption due to printed BP, we attribute the forward photo detection ability to BP induced charge transfer, a photo-gating effect previously observed in graphene-based hybrid devices[69, 70]. This may alter the Fermi energy of graphene and consequently lead to variations in the Gr/Si Schottky barrier height that controls the charge transport, as shown in inset of Fig. 5e. BP/Gr/Si exhibits a ~0.55 ms response (Supplementary Fig. 16), and a long-term (> 7 days; Supplementary Fig. 17) operation stability under ambient. Supplementary Table 3 presents the comparison between our work and the current BP photodetectors with different device structures reported thus far. Our visible to near-infrared BP photodetector operating in both forward and reverse configurations demonstrates the potential of printed BP in high performance broadband optoelectronic applications. We note that using our BP ink, there is significant potential to integrate with alternative device structures including CMOS compatible and waveguide-based near-infrared photodetectors for silicon photonics[63].

## Discussion

In summary, we have formulated a binder-free, two-component functional BP ink for inkjet printing, producing uniform material deposition on untreated substrates. Among the three organic solvents we investigate for UALPE, through AFM, optical absorption and light scattering, we demonstrate that NMP is the most effective solvent for BP exfoliation. Since the NMP dispersion is poorly suited to inkjet printing, we use a solvent exchange process to formulate a BP ink in a binary solvent carrier. The ink ensures stable jetting, appropriate wetting, minimised coffee ring effect and rapid ink drying (< 10 s) at low temperatures (< 60 °C). By examining the droplet spacing and substrate temperature, we determine the best printing conditions for the ink to allow high printing consistency (< 2% variation) and spatial uniformity (< 3.4% variation). We show that encapsulating the printed BP with a parylene-C passivation layer can preserve it against oxidation for > 30 days. We then demonstrate a highly stable SA for ultrafast lasers and a broadband photodetector with enhanced responsivity using our BP ink. With the printed BP-SA, we demonstrate the generation of very stable ultrashort pulses under an intense irradiation of 32.7 MW cm$^{-2}$ for over 30 days. The photodetector not only shows > 10 times enhancement in detection performance at 450 nm, but also extends the detection range to include 1550 nm. Our BP ink formulation and the demonstrated devices highlight the potential for printed BP in a broad range of long-term stable optoelectronic and photonic systems even when operating under ambient conditions.

## Methods

**Production and characterisation of dispersions**. Bulk BP crystals (Smart Elements) are mixed into anhydrous NMP, CHP and IPA (Sigma Aldrich) in sonication tubes at an initial concentration of 1 gL$^{-1}$ under ambient conditions. The tubes are then backfilled with nitrogen and sealed. The mixtures are sonicated for 12 h in a 20 kHz bath sonicator at 15 °C. The resultant dispersions are loaded into tubes, backfilled with nitrogen and centrifuged for 30 min to sediment the unexfoliated flakes. The centrifuge speeds chosen are 1, 2, 3 and 4 krpm (equivalent to 95, 380, 850 and 1500 × g, respectively). The upper 80% of resultant dispersions is collected for analysis and ink formulation. The concentration of dispersed BP is estimated from the optical extinction at 660 nm via Beer–Lambert law. The dispersions are diluted to 10 vol% to avoid detector saturation. The sample for Raman measurement of exfoliated BP flakes in NMP is prepared by drop-casting the NMP dispersion onto a Si/SiO$_2$ and subsequently dried at 70 °C under nitrogen. Raman characterisation of the BP uses an excitation wavelength of 514 nm with a power < 0.1 mW and a duration of 10 s for each measurement point.

**Ink formulation and printing**. A NMP-based dispersion of BP is prepared as above. A secondary centrifugation step is then used (275,000 × g for 30 min) to sediment the dispersed BP. Next, NMP is removed and the sedimented BP flakes are then redispersed in IPA through sonication for 10 min. The volume of IPA used is 10% of the removed NMP, allowing the concentration of dispersed BP to be increased. 2-butanol (10 vol%) is then added to formulate the ink. The ink is characterised via pendant droplet and parallel plate rheometer measurements to determine the surface tension and the viscosity of the ink, respectively. For contact angle measurement, a ~2 μL droplet is dropcast and measured via sessile drop technique. The measurement is conducted at room temperature (~20 °C). The inkjet printer used in this work is a Fujifilm Dimatix DMP-2831, and the cartridge is Dimatix DMC-11610 which produces ~10 pL droplets. The substrates, including Si/SiO$_2$ (SiO$_2$ thickness-100 nm), glass and PET, are cleaned with acetone/IPA/DI water prior to printing. No other surface treatment is used.

**Data availability**. The data that support the findings of this study are available from the corresponding authors upon request.

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

## Acknowledgements

We thank I. Goykhman for useful discussions, and C. George and M. De Volder for access to the scattering measurement system. T.A.-O. and R.C.T.H. acknowledge funding from EPSRC grant EP/L016087/1 and EP/G037221/1, K.S. from National Science Foundation China (NSFC, No. 51650110494), R.I.W. from EPSRC doctoral prize fellowship, H.J. from Jenny ja Antti Wihuri-foundation, Z.S. from European Unions Seventh Framework Programme (REA grant agreement No. 631610), Academy of Finland (No. 276376), and TEKES (OPEC), M.Z. from Beihang University through Zuoyue program and funding from 973 Program (2012CB315601). Y.X. from ZJ-NSF (LZ17F040001) and Fundamental Research Funds for the Central Universities (2016XZZX001-05, 2017XZZX008-06, 2017XZZX009-02), M.Z., Y.X. and P.T. from NSFC (Nos. 61521091, 61505005, 61435002, 61274123, 61674127, 61474099, 11434010 and 11474277), Y.X. and P.T. from the National Key Research and Development Program of China (2016YFA0200204, 2016YFA0301204), E.J.R.K. and T.H. from Royal Academy of Engineering (RAEng) Research Fellowships. The ultrathin PET (1.5 μm) substrate is provided by DuPont Teijin Films.

## Author contributions

G.H., R.C.T.H., Z.S., M.Z., X.Y. and T.H. designed the experiments. G.H., T.A.-O., X.J., A.A., Y.H., R.C.T.H., K.S., Z.Y., X.Z. and H.J. performed the experiments. G.H., R.I.W., T.-C.W., J.-B.W., P.H.T., Z.S., M.Z., Y.X. and T.H. analysed the data. G.H., R.I.W. and E.J.R.K. prepared the figures. G.H., R.I.W., P.H.T., E.J.R.K., M.Z., Y.X. and T.H. wrote the manuscript. All authors discussed results from the experiments and commented on the manuscript.

## Additional information

**Competing interests:** The authors declare no competing financial interests.

