## [Peer Review File · Nature Communications]

Reviewers' Comments:

Reviewer #1 (Remarks to the Author)

In this work Hu et al show a nice work on formulation engineering of phosphorene-based inks. The exfoliation is performed in NMP, a traditional solvent used for bath-sonication assisted exfoliation, and then a solvent exchange is performed in IPA, and 2-butanol. This combination of solvents allows for fast drying and reduction of the coffee ring, typically observed for NMP. The ink obtained after solvent exchange is printable, stable and highly concentrated, allowing for fast printing. Lateral size and thickness of the flakes is analysed by AFM.

It is well known that phosphorene, in the form of single and few layers, is strongly hygroscopic, leading to reduced stability in air due to oxidation. The authors use Raman spectroscopy to analyse the stability of the material in solution and as printed, after encapsulation with parylene. Based on these measurements, the authors claim the material not to be oxidised in solution and to get slightly oxidised once encapsulated. They finally use encapsulated printed phosphorene as saturable absorber and in gr/si photodetectors, showing better performance compared to the case without phosphorene.

The results presented by the authors are interesting, as up to now nobody ever reported printable inks formulation of phosphorene, therefore this is an important result, which could be of great interest for the community. However, there are some weak points with this work:

- the printable formulation, since based on IPA as solvent, will have issues related to toxicity, so this may limit applications. The authors should comment on this.
- is the water signal in the uv-vis spectrum normally observed also with other solvents, such as NMP? where the water is coming from? This may be related to the source of oxidation observed by raman.
- towards the end of pag3 - based on the position of the peaks, the authors cannot comment on the crystallinity of the material, which affects more the FWHM of the Raman peaks than their positions. the match in the Raman positions indicates that they have indeed a phosphorene-like material, but they cannot say anything about crystallinity.
- To the best of my knowledge ref 20 only shows that the Raman intensity decreases for longer exposure to the air. The threshold suggested by the authors strongly depends on the experimental conditions, as a Raman peak is simply not seen when its intensity is below the noise or sensitivity of the spectrometer. I do not think this method is reliable to evaluate the degree of oxidation of the material and I would suggest to remove it. The intensity ratio between raman peaks also has error bars, typically around 10-20%, so it would be difficult to conclude something reliable based on those data.
- similarly, because external effects can contribute to changes in the Raman intensity ratio, I would suggest also to remove the Raman analysis of the encapsulated Ph-ene (if the authors really want to keep it, then please move everything in the SI). XPS would be the best technique to conclude on stability, but the sample will need to be transferred in the chamber as soon as it is printed as the encapsulation layer cannot be used.
- concerning the devices, it would have been nice to see a novel device based on this material, such as a transistor or a fully printed photodetector as in ref 15. However, I do understand that the paper focuses more on the formulation optimisation for ink-jet printing and hopefully novel devices will be proposed in future works. Still, some revisions are needed also for the devices' results, as following:

a) Dark current should be given for Figure 5 f and g, maybe in supplementary to truly assess performance. If the dark current is 1A but there is only an increase of 10uA the performance is not good.

b) The device without printed Ph-ene is also photoactive, as seen in Fig. 5 f and g. This data without BP should be scaled up and included in SI to assess the improvement in performance with the BP layer. Also, the dark current curves should be provided for these data sets.

c) The laser power of $\sim 20\text{mW}$ used in Fig 5g is quite high compared to what has been reported in literature. The power density and the size of the photoactive region should be reported.

d) These photodetectors usually have quite slow photo-response so it would be good to see some data showing the photocurrent produced over time switching on/off the laser.

e) measurements as a function of the air exposure time should also be provided to confirm stability (see points above).

Reviewer #2 (Remarks to the Author)

The manuscript by Hu et al. reports an inkjet printing process for liquid exfoliated BP. To my knowledge, inkjet printing of BP has not yet been demonstrated. On the downside, there are some major weaknesses in the results and analysis that need to be addressed before any recommendation can be made:

The Raman intensity ratio of $I(\text{A1g})/I(\text{A2g})$ have seen to be polarization dependent in several reports other than the paper by Favron et al.; examples are: i) Anisotropic Electron-Photon and Electron-Phonon Interactions in Black Phosphorus, ii) Rediscovering black phosphorus as an anisotropic layered material for optoelectronics and electronics, and iii) High-Quality Black Phosphorus Atomic Layers by Liquid-Phase Exfoliation. In these reports, although A1g and A2g follow a similar trend, but the ratio is indeed polarization dependent, and in some angles can easily exceed 1. Favron et al. have shown that A1g peak diminishes earlier than A2g peak during the oxidation which is mainly due to the nature of degradation in the BP and its effect on the crystal vibrational modes. Although they have also reported a very rough insensitivity for the $I(\text{A1g})/I(\text{A2g})$ to the polarization, but in the light of the other reports, polarization insensitivity needs to be independently verified by the authors before making any conclusion on the oxidation.

The AFM images shown in Fig. 2g are extremely low mag and the height scale bar is chosen so small that the data falls mainly outside of the bounds and show saturation (black or white). These figures are not informative regarding the uniformity of the printing. The comparison to NMP is also not very informative. It is well known and trivial that high boiling temperature solvents cannot be directly used for printing. In general, the use of alcohols as suitable solvents for inkjet printing is also demonstrated, as pointed out by the authors. However, stability of the BP dispersions has neither been reported before, nor characterized by the authors. If BP flakes re-aggregate in IPA, the ink would not be viable. I suggest the authors check the optical absorption coefficients of BP in IPA after a long sitting time.

Here are some minor comments/questions about the manuscript:

1. What was the real time difference between the three cases of drying of droplets while measuring the contact angle?
2. What was the effect of the process on blockage of nozzle? If clogging happened, how soon it happened and what is the potential solution?
3. How did the process of adding secondary alcohol, 2-butanol, influence the solution concentration? How did the authors estimate the final concentration of the solution after the last step of solvent addition?
4. How did authors find the optimized amount of secondary alcohol, 2-butanol? How the amount of 2-butanol affects the surface tension gradient of the jetted droplets and coffee ring residues?
5. How different was the result on hydrophilic and hydrophobic substrates? (Si or SiO_2)
6. In the second page of the manuscript, in the last paragraph before results section, it is stated that, "In addition, the low boiling point of alcohols leads to their fast evaporation and rapid ink drying ($<10\text{ s}$ at $<60\text{ }^\circ\text{C}$) without requiring high temperature curing. This combination of few printing repetitions and rapid drying ensures that the time for BP oxidation during the printing process significantly minimized." The last sentence is somehow ambiguous. In a way it seems to

convey that, as the time of oxidation is minimized, BP oxidizes faster which is not the case. The faster drying process, provides low possibilities of BP oxidation.

Reviewer #3 (Remarks to the Author)

This paper presents details of the production of black phosphorous (bP) nano-sheets by liquid-phase exfoliation, and the formulation into inks suitable for ink-jet printing. The authors show that the printed black phosphorous can be stabilised against oxidation by encapsulation, over time periods of up to 30 days. These inks have then been used to produce demonstrator devices in optical applications, namely mode-locked laser and photodetector. While liquid-phase exfoliation of bP has been shown before, the optimisation of the dispersions to inkjet printing applications is novel, and will be of interest to researchers working in the field of 2D materials and printed electronics. Importantly, it demonstrates that liquid-phase exfoliation can be used to produce such devices, which is important for potential industrial applications and scale-up.

The manuscript is well written and clearly explains the experimental work and the implications of the results generated. However, the following points should be addressed to further strengthen the paper, and widen the potential audience.

1. The results of the optical absorption spectroscopy are shown as the extinction and scattering spectra, and a peak is claimed in the extinction spectrum. This peak is not clearly visible, however I suspect that it will be more visible in the pure absorption spectrum, and as such I would like to see this presented. This would potentially also give an indication of the band gap of the exfoliated material. Given samples have been measured in an integrating sphere spectrometer, this data should have already been gathered.

2. The use of the scattering spectra to characterise the flake size is a useful approach. It would be useful to apply this technique to the dispersions after solvent transfer to the IPA/butanol mixture to probe for any flake aggregation before printing. Also, do the results here agree with the scattering exponent to length fit presented by Hanlon et al (Nat. Comm. 2015)? And if not, is there an explanation for the discrepancy?

3. The discussion of the raman maps in fig. 2e and 4d state that "there are no localised regions with [intensity ratio] > 0.6." However both the maps and corresponding histograms show that there are locations with a ratio over 0.6. The authors should clarify this apparent discrepancy. Also, although the cited paper (Favron et al.) shows that ratios lower than 0.2 are indicative of higher levels of oxidation, there is no evidence shown there that values above 0.6 also indicate oxidation. The present authors claim that values >0.6 indicate oxidations, and so a clearer explanation of this is needed. Perhaps XPS measurements would confirm degree of oxidation that can then be correlated to the raman measurements.

4. Did the authors attempt to exfoliate directly in the IPA/Butanol mixture that was used for inkjet printing? Given the resulting dispersions are stated as being stable against sedimentation for weeks, I would be interested to know if this was attempted, and if so, how it compared to the other solvents. Avoiding the solvent exchange step would make the process more attractive to commercial scale-up.

5. Can the authors clarify the technique used for the contact angle measurements? The methods state a pendant drop technique, but the discussion on p. 5, paragraph 3 talks about deposition of material at the droplet edge, and contact line sticking. This suggest that sessile drop approach has been used.

6. Fig. 4a shows a noticeable deviation from the linear trend for the first and second printing passes. Do the authors have an explanation for this?

7. The study on the stability of printed material, with and without encapsulation is clearly important, and highly relevant for potential users of this material and techniques. What would be

equally important, and would improve the relevance of the paper would be equivalent measurement of the inks themselves. This would allow an estimation of a shelf-life for inks, which may be produced separate from the printing location.

8. The performance over time of the saturable absorber has been demonstrated, and so it would be interesting to see similar results from the photodetector. For those not familiar with the specific applications demonstrated, it would be helpful to include, if possible, a comparison with current technologies, and/or a discussion of the advantages that inkjet black phosphorous based devices offer. This would help to widen the potential audience of the paper.

Point by Point Response to the Reviewers' Comments

Reviewer #1

In this work Hu et al show a nice work on formulation engineering of phosphorene-based inks. The exfoliation is performed in NMP, a traditional solvent used for bath-sonication assisted exfoliation, and then a solvent exchange is performed in IPA, and 2-butanol. This combination of solvents allows for fast drying and reduction of the coffee ring, typically observed for NMP. The ink obtained after solvent exchange is printable, stable and highly concentrated, allowing for fast printing. Lateral size and thickness of the flakes is analysed by AFM.

It is well known that phosphorene, in the form of single and few layers, is strongly hygroscopic, leading to reduced stability in air due to oxidation. The authors use Raman spectroscopy to analyse the stability of the material in solution and as printed, after encapsulation with parylene. Based on these measurements, the authors claim the material not to be oxidised in solution and to get slightly oxidised once encapsulated. They finally use encapsulated printed phosphorene as saturable absorber and in gr/si photodetectors, showing better performance compared to the case without phosphorene.

The results presented by the authors are interesting, as up to now nobody ever reported printable inks formulation of phosphorene, therefore this is an important result, which could be of great interest for the community. However, there are some weak points with this work:

1. the printable formulation, since based on IPA as solvent, will have issues related to toxicity, so this may limit applications. The authors should comment on this.

We thank the reviewer for this very important comment. While we note that IPA does have a level of toxicity, it is comparably benign when compared to the solvents widely used for inkjet ink formulations containing 2d materials, including graphene and transition metal dichalcogenides, which are typically based on harsh, toxic solvents such as N-Methyl-2-pyrrolidone (NMP).^{1,2} Such organic solvents are not only demonstrably harmful for human health and the environment, but can also be incompatible with many polymeric substrates, limiting the applications of these inks. It is, in part, specifically to avoid using these toxic solvents, that we have formulated the ink using IPA. Quoting the safety data sheet provided by Sigma Aldrich, the acute toxicity limits for IPA are oral - 5,045 mg/kg (NMP: 3,914 mg/kg), inhalation - 16000 ppm (NMP: 5100 ppm) and dermal - 12,800 mg/kg (NMP: 8,000 mg/kg), and the UK workplace exposure limit is 500 ppm (NMP: 10 ppm). Given its relatively low hazardous potential, IPA is widely used in rubbing alcohol, hand sanitisers and household cleaning products, as well as various commercial functional and pigment-based inks. For instance, IPA is used as a solvent component in commercial silver nanoparticle inkjet inks. IPA will also not cause damage to commonly used polymeric substrates, making it compatible for the development of flexible, printed devices. In addition, as we have demonstrated in the manuscript, IPA enables an ink formulation with high loading ($\sim 5 \text{ gL}^{-1}$), stable single-droplet jetting, and appropriate wetting of the substrates. The low boiling point of IPA (82.6°C) also allows a rapid ink drying (<10 s) at low temperatures (<60°C). This rapid ink drying, in combination with the high ink loading, leads to a significantly reduced printing time, giving low possibilities for BP oxidation. This is of vital importance for inkjet printing of BP.

In conclusion, we acknowledge, as the reviewer suggests, that there are mild drawbacks to the use of IPA as a solvent. However, since a variety of commercial inks for different printing systems contain IPA due to its low hazardous potential, and because of the suitability and the necessity of using IPA as the ink carrier for printing of BP, we believe IPA is the best solvent for our inks.

To clarify this, we have made the following changes in the revised manuscript:

Indeed, alcohols are commonly used as one of the carrier solvents in graphics and functional inks. → **Regarding alcohols, given their relatively low hazardous potential, alcohols such as IPA are widely used in various commercial functional and pigment-based inks.**

We have also included this discussion in Supplementary Note 3:

IPA is used as the major ink carrier solvent for ink formulation in our work. Though we acknowledge that IPA does have a level of toxicity, it is comparably benign when compared to the solvents widely used for inkjet ink formulations containing 2d materials, including graphene and transition metal dichalcogenides, which are typically based on harsh, toxic solvents such as N-Methyl-2-pyrrolidone (NMP). Such organic solvents are not only demonstrably harmful for human health and the environment, but can also be incompatible with many polymeric substrates, limiting the applications of these inks. It is, in part, specifically to avoid using these toxic solvents, that we are formulating the ink using IPA. Quoting the safety data sheet provided

by Sigma Aldrich, the acute toxicity limits for IPA are oral - 5,045 mg/kg (NMP: 3,914 mg/kg), inhalation - 16000 ppm (NMP: 5100 ppm) and dermal - 12,800 mg/kg (NMP: 8,000 mg/kg), and the UK workplace exposure limit is 500 ppm (NMP: 10 ppm). Given its relatively low hazardous potential, IPA is widely used in various commercial functional and pigment-based inks, *e.g.* commercial silver nanoparticle inkjet inks. IPA will also not cause damage to commonly used polymeric substrates, making it compatible for the development of flexible, printed devices. In addition, as we have demonstrated in the manuscript, IPA is possible to formulate an ink for high loading ($\sim 5 \text{ gL}^{-1}$), stable single-droplet jetting, and appropriate wetting of the substrates. The low boiling point of IPA (82.6°C) can also allow a rapid ink drying ($< 10 \text{ s}$) at low temperatures ($< 60^\circ\text{C}$). This rapid ink drying, in combination with the high ink loading, leads to a significantly reduced printing time, giving low possibilities for BP oxidation. This is of vital importance for inkjet printing of BP.

2. Is the water signal in the uv-vis spectrum normally observed also with other solvents, such as NMP? where the water is coming from? This may be related to the source of oxidation observed by raman.

We thank the reviewer for this question. We do indeed observe water signals with all the solvents (NMP, N-Cyclohexyl-2-pyrrolidone (CHP) and IPA) used for the exfoliation of black phosphorus (BP) in the near infrared wavelength region,^{3,4} *i.e.* the peaks at $\sim 0.91 \mu\text{m}$, $\sim 1.01 \mu\text{m}$, $\sim 1.19 \mu\text{m}$ and $\sim 1.39 \mu\text{m}$; Fig. R1. These signals demonstrate that there is trace water in these solvents. We note that the solvents used in our experiments were purchased in anhydrous composition, with the aim of minimising water content in the solutions. Assuming the quality of the chemicals bought from the supplier is as quoted, we suggest that the measured water signals indicate that moisture may have been introduced into the solvents during the absorbance measurement itself. Indeed, given the solvents are hygroscopic, ambient humidity absorption is inevitable to some degree, and may have contributed to the $\sim 1\%$ increase in oxidation between bulk and exfoliated BP.

Figure R1. Optical absorbance (log-log scale) of the anhydrous NMP, CHP and IPA for BP exfoliation.

We have made the following changes in the revised manuscript:

We observe variations at ~ 1.2 and $1.4 \mu\text{m}$. We attribute these variations to the water molecules absorbed by the solvents during the exfoliation process and the absorbance measurement. \rightarrow **We attribute the variations observed at the near-infrared region to the ambient moisture absorbed by the solvents during the exfoliation process; see Supplementary Fig. 1(a) and associated discussion.**

We have also included Fig. R1 as Supplementary Fig. 1(a) with discussion in Supplementary Note 1:

Supplementary Fig. 1(a) presents the optical absorbance spectra of the solvents (N-Methyl-2-pyrrolidone (NMP), N-Cyclohexyl-2-pyrrolidone (CHP) and isopropanol (IPA)) used for BP exfoliation. As observed, there are water signals with all the three solvents in the near infrared wavelength region, *i.e.* the peaks at $\sim 0.91 \mu\text{m}$, $\sim 1.01 \mu\text{m}$, $\sim 1.19 \mu\text{m}$ and $\sim 1.39 \mu\text{m}$. These signals demonstrate that there is trace water in these solvents. We note that the solvents used in our experiments were purchased in anhydrous composition, with the aim of minimising water content in the solutions. Assuming the quality of the chemicals bought from the supplier is as quoted, we suggest that the measured water signals indicate that moisture may have been introduced into the solvents during the handling for absorbance measurement. Indeed, given the solvents are hygroscopic, ambient humidity absorption is inevitable to some degree, and may have contributed to the $\sim 1\%$ increase in oxidation between bulk and exfoliated BP.

3. towards the end of pag3 - based on the position of the peaks, the authors cannot comment on the crystallinity of the material, which affects more the FWHM of the Raman peaks than their positions. the match in the Raman positions indicates that they have indeed a phosphorene-like material, but they cannot say anything about crystallinity.

We thank the reviewer for this comment. Regarding the peak positions, to quote page 4 of “Isolation and characterization of few-layer black phosphorus⁵”: “The peaks... are due to vibrations of the crystalline lattice of the black phosphorus...” Therefore, we state in our manuscript, “This suggests that our exfoliated BP flakes are highly crystalline.”

To further clarify this statement, we have made the following changes in the revised manuscript:

This suggests that our exfoliated BP flakes are highly crystalline. → **Since the peaks are due to BP crystalline lattice vibrations, this consistency suggests that our exfoliated BP flakes are highly crystalline.**

Regarding the peak FWHM, we quote the Fig. S13 (a) of “Photooxidation and quantum confinement effects in exfoliated black phosphorus” - Ref. 6 in this response; Fig. R2(a). As shown, the FWHM are all within $\sim 2\text{-}6.5\text{ cm}^{-1}$ (A_g^1), $\sim 2\text{-}7\text{ cm}^{-1}$ (B_{2g}), and $\sim 2\text{-}8\text{ cm}^{-1}$ (A_g^2) for 1-6 layer/s of mechanically exfoliated and bulk BP. In Fig. R2(b), we show that the typical FWHM of our exfoliated and bulk BP samples (quoting Fig. 1(d) in the revised manuscript) are consistent with those in literature. We further acquire the FWHM statistics from the Raman mapping measurements (~ 360 measurement points) of exfoliated and bulk BP, and show that the FWHM statistics are also consistent with those reported in literature; Fig. R2(c).

Figure R2. Full-width at half maximum (FWHM) of BP Raman: (a) adapted from Fig. S13 (a) of “Photooxidation and quantum confinement effects in exfoliated black phosphorus” - Ref. 6 in this response; (b) typical Raman spectrum with FWHM for the exfoliated and bulk BP; (c) FWHM statistics.

We have included the following additional text in the revised manuscript:

Statistical investigation (~ 360 measurements) on the full width at half maximum (FWHM) of these three peaks also notes no discrepancy from those reported in literature (Supplementary Fig. 5), further suggesting high crystallinity.

We have also included Fig. R2(b,c) as Supplementary Fig. 5 with discussion in Supplementary note 2:

Preceding studies show the full-width at half maximum (FWHM) are all within $\sim 2\text{-}6.5\text{ cm}^{-1}$ (A_g^1), $\sim 2\text{-}7\text{ cm}^{-1}$ (B_{2g}), and $\sim 2\text{-}8\text{ cm}^{-1}$ (A_g^2) for 1-6 layer of mechanically exfoliated and bulk BP. In Supplementary Fig. 5(a) (*i.e.* Fig. 1(d) with FWHM labels), we show that the typical FWHM of our exfoliated and bulk BP samples are consistent with those in literature. We further acquire the FWHM statistics from the Raman mapping measurements (~ 360 measurement points) of exfoliated and bulk BP; Supplementary Fig. 5(b). We demonstrate that the FWHM statistics are also consistent with those in literature. The consistency in the FWHM suggests the crystallinity of our exfoliated BP.

4. To the best of my knowledge ref 20 only shows that the Raman intensity decreases for longer exposure to the air. The threshold suggested by the authors strongly depends on the experimental conditions, as a Raman peak is simply not seen when its intensity is below the noise or sensitivity of the spectrometer. I do not think this method is reliable to evaluate the degree of oxidation of the material and I would suggest to remove it. The intensity ratio between raman peaks also has error bars, typically around 10-20%, so it would be difficult to conclude something reliable based on those data.

We thank the reviewer for the suggestion. We acknowledge that the intensity of the Raman peaks is susceptible to the experimental conditions, and as such, the intensity may be below the noise or sensitivity level of the spectrometer. Indeed, as indicated by the grey areas in the mapping (consisting of 20×20 data points) - Fig. R3(a) (Fig. 1(e) in the manuscript), we find that the Raman intensity of 10% of the data points is not strong enough for interpretation. We plot a representative Raman spectrum of this type as #1, Fig. R3(b). These low intensity data points are discarded for BP oxidation investigation. We also show a representative Raman spectrum with $I(A_g^1)/I(A_g^2)$ within the threshold 0.2-0.6 as #2, and outside as #3, respectively; Fig. R3(b). These remaining (~ 360) data points have strong Raman intensity for BP oxidation investigation. This method therefore is, we believe, reliable for oxidation investigation.

Figure R3. (a) Raman map of the intensity ratio, $I(A_g^1)/I(A_g^2)$, with $1 \mu\text{m}$ spatial step. The grey squares correspond to regions where the Raman intensity is too low for accurate interpretation; (b) Raman spectra of #1, #2 and #3.

We draw our conclusions on the oxidation based on the statistical distribution of these ~ 360 intensity ratios across the sample, which should be sufficient to negate any errors in the peak intensities at individual points. We present the $I(A_g^1)/I(A_g^2)$ values as in mapping and histogram forms in the submission, therefore we do not present the error bars.

In conclusion, we believe that our protocol is sufficiently reliable to indicate the oxidation in the samples.

To clarify how the Raman data is processed, we have included Fig. R3 as Supplementary Fig. 4 and the discussion in Supplementary Note 2:

The Raman peaks can be susceptible to the experimental conditions, and as such, the intensity may be below the noise or sensitivity level of the spectrometer. Indeed, as indicated by the grey areas in the Raman mapping (consisting of 20×20 data points) - Supplementary Fig. 4(a) (see Fig. 1(e)), we find that the Raman intensity of 10% of the data points is not strong enough for interpretation. We plot a representative Raman spectrum of this type as #1, Supplementary Fig. 4(b). These low intensity data points are discarded for BP oxidation investigation. We also show a representative Raman spectrum with $I(A_g^1)/I(A_g^2)$ within the threshold 0.2-0.6 as #2, and outside as #3, respectively; Supplementary Fig. 4(b). These remaining (~ 360) data points have strong Raman intensity for BP oxidation investigation. In conclusion, we feel that our protocol is sufficiently reliable for oxidation investigation.

5. similarly, because external effects can contribute to changes in the Raman intensity ratio, I would suggest also to remove the Raman analysis of the encapsulated Ph-ene (if the authors really want to keep it, then please move everything in the SI). XPS would be the best technique to conclude on stability, but the sample will need to be transferred in the chamber as soon as it is printed as the encapsulation layer cannot be used.

We thank the reviewer for the suggestion. We acknowledge that the external factors can affect the Raman intensity ratio, $I(A_g^1)/I(A_g^2)$ (and indeed state this in the manuscript while discussing the Raman analysis of parylene-C encapsulation, quoting:

“We suggest that this is due to interactions between BP and parylene-C”, and “We therefore propose that this sizeable increase in $I(A_g^1)/I(A_g^2)$ in our printed sample after encapsulation is partially as a result of the increase in $I(A_g^1)$...”). We demonstrate that parylene-C encapsulation causes changes in the ratio even for freshly cleaved bulk BP, where significant oxidation is unlikely. However, our intention here is to demonstrate the stability of the intensity ratio in the sample following encapsulation over a period of days, rather than the ratio itself against oxidation. We also confirm the stability against oxidation *via* optical extinction, Fig. 4 (e) in the manuscript, by showing that encapsulation of BP with parylene-C protects it from oxidation and stabilises the sample for over 30 days. In showing this, we aim to highlight that while external effects play a significant role on the ratio (as has been clearly highlighted by us and Favron *et al.*⁶), the changes in $I(A_g^1)/I(A_g^2)$ on parylene-C encapsulation are not solely (if at all) related to degradation by oxidation. We believe that we have now clarified our position and would like to keep the relevant data in the main manuscript as it adds an important aspect to our manuscript.

Finally, we fully agree with the reviewer regarding the use of XPS. While XPS is indeed a very reliable technique to study BP oxidation, it would not allow us to understand the stabilising effect of a parylene-C encapsulation layer, which is the main aim of this particular section of the manuscript.

6. Concerning the devices, it would have been nice to see a novel device based on this material, such as a transistor or a fully printed photodetector as in ref 15. However, I do understand that the paper focuses more on the formulation optimisation for ink-jet printing and hopefully novel devices will be proposed in future works. Still, some revisions are needed also for the devices’ results, as following:

- a) Dark current should be given for Figure 5 f and g, maybe in supplementary to truly assess performance. If the dark current is 1A but there is only an increase of 10uA the performance is not good.
- b) The device without printed Ph-ene is also photoactive, as seen in Fig. 5 f and g. This data without BP should be scaled up and included in SI to assess the improvement in performance with the BP layer. Also, the dark current curves should be provided for these data sets.

We thank the reviewer for the comments. As the reviewer has mentioned, the primary aim of this work has indeed been the formulation of BP ink for inkjet printing. Having optimised the ink formulation, we investigated its potential in fabrication of optoelectronic and photonic devices *via* inkjet printing. Besides the demonstration of saturable absorbers (SA), we consider the development of a photodetector to be a promising avenue, given BP’s direct bandgap of 0.3-2.0 eV, spanning the visible and infrared wavelength range. The hybrid structure BP/Gr/Si Schottky junction photodetector included in our manuscript is just one such application which highlights the potential of our ink for broadband optoelectronics.

We have attached the dark currents of Gr/Si and BP/Gr/Si in the same figure on log-log scale for a better comparison; Fig. R4. As shown, the dark current of Gr/Si when reverse biased is on the ~ 1 nA scale, while it is ~ 100 nA for BP/Gr/Si. This could arise from the doping of the CVD graphene by the printed BP layer as we have discussed in the manuscript, and may have been responsible for the improvement in the device performance in terms of photocurrent change. Since here we are here studying the effect of printed BP on device performance, we therefore suggest not to scale up the photocurrent changes of the Gr/Si to compare those of the BP/Gr/Si.

Figure R4. Dark current of Gr/Si and BP/Gr/Si.

As we can observe, indeed the forward photocurrent changes are small (up to ~ 7.5 μ A, quoting Fig. 5 (f, g) in the manuscript) compared to the dark currents (up to ~ 900 μ A). However, the reverse photocurrent changes are on μ A scale (up to ~ 12.5 μ A,

quoting Fig. 5 (f, g)), significantly different from the dark currents (on a nA scale, up to ~ 100 nA). This demonstrates the printed BP layer does indeed lead to a device performance improvement. In showing this, we aim to demonstrate the potential of our BP ink for printed optoelectronics and photonics. We are currently working on the optimisation of this device structure to further improve the device performance.

We are also developing further printed devices with the BP ink. One such device is a vertically structured photodetection diode, fabricated by inkjet printing BP directly onto Si, as shown in Fig. R5(a). Here we attach the device photo response under 450 nm illumination; Fig. R5(b). This demonstrates the device is capable of photodetection. However, we are still investigating the physical mechanisms behind the device operation, and optimising the device structure. Such investigation falls outside the scope of this work and is not included in the manuscript.

Figure R5. (a) BP/Si photodetector structure; (b) Photocurrent of printed BP/Si photodetector.

We have included Fig. R4 as Supplementary Fig. 16(a) and the discussion in Supplementary Note 8:

Supplementary Fig. 16(a) presents the dark currents of Gr/Si and BP/Gr/Si in the same figure on log-log scale for a better comparison. As shown, the dark current of Gr/Si when reversed biased is on the ~ 1 nA scale, while it is ~ 100 nA for BP/Gr/Si. This could arise from the doping of the CVD graphene by the printed BP discussed in the manuscript, and may have been responsible for the improvement in the device performance in terms of photocurrent change. Since we are here studying the effect of printed BP on device performance, we therefore do not scale up the photocurrent changes of the Gr/Si to compare those of the BP/Gr/Si.

As we can observe, indeed the forward photocurrent changes are small (up to ~ 7.5 μ A; Fig. 5(f, g)) compared to the dark currents (up to ~ 900 μ A). However, the reverse photocurrent changes are on the μ A scale (up to ~ 12.5 μ A; Fig. 5 (f, g)), significantly different from the dark currents (on the nA scale, up to ~ 100 nA). This demonstrates the printed BP layer does indeed lead to a device performance improvement. In showing this, we aim to discuss the potential of our BP ink for printed optoelectronics and photonics.

7. c) The laser power of ~ 20 mW used in Fig 5g is quite high compared to what has been reported in literature. The power density and the size of the photoactive region should be reported. This needs to be updated.

We thank the reviewer for the observation. In our original manuscript, we reported the laser output power: 2.38, 256.8, 513 and 1200 μ W for 450 nm, and 5.9, 10.6, 15.28 and 19.9 mW for 1550 nm, respectively. A survey of the literature shows that the intensity at 450 nm (up to 1.2 mW) is comparable to that used for similar devices, for instance photodetector based on mechanically exfoliated BP (up to 1.9 mW⁷) and inkjet-printed other 2d materials (up to 2 mW⁸). The 1550 nm intensity (up to 20 mW) is indeed higher than those mentioned above. This is because BP has a much lower absorbance for infrared light, quoting Fig. 1(b) in the manuscript.

The laser beam diameter is ~ 2 mm, giving a beam spot area as ~ 3.14 mm². The power densities are therefore estimated as 0.76, 81.78, 163.38 and 382.17 W/m² for 450 nm, and 1.88×10^3 , 3.38×10^3 , 4.87×10^3 and 6.34×10^3 W/m² for 1550 nm, respectively. The photoactive region of our photodetector is 450 μ m \times 450 μ m. Therefore, the actual incident laser powers on the photoactive region are 0.15, 16.56, 33.08 and 79.11 μ W for 450 nm, and 0.38, 0.68, 0.99 and 1.28 mW for 1550 nm, respectively. These power values are smaller than those in the cited works above.

Based on the suggestion from the reviewer, we have updated the laser power and the corresponding photodetector responsivity in our revised manuscript, and have included the measurement details in Supplementary Note 8:

We measure the devices with four different laser output powers for each wavelength in our work, which are 2.38, 256.8, 513 and 1200 μW for 450 nm, and 5.9, 10.6, 15.28 and 19.9 mW for 1550 nm, respectively. The laser beam diameter is ~ 2 mm, giving a beam spot area of 3.14 mm^2 . The power densities are therefore estimated as 0.76, 81.78, 163.38 and 382.17 W/m^2 for 450 nm, and 1.88×10^3 , 3.38×10^3 , 4.87×10^3 and 6.34×10^3 W/m^2 for 1550 nm, respectively. The photoactive region of our photodetector is $450 \mu\text{m} \times 450 \mu\text{m}$. Therefore, the actual incident laser powers on the photoactive region are 0.15, 16.56, 33.08 and 79.11 μW for 450 nm, and 0.38, 0.68, 0.99 and 1.28 mW for 1550 nm, respectively.

8. d) These photodetectors usually have quite slow photo-response so it would be good to see some data showing the photocurrent produced over time switching on/off the laser.

We thank the reviewer for this suggestion. Here we attach the time response of Gr/Si and BP/Gr/Si at 1550 nm; Fig. R6(a). Gr/Si shows no response, whereas BP/Gr/Si exhibits a stable and reproducible response. The response time of the cycle shown in Fig. R6(b) is ~ 0.55 ms, and the recovery time is ~ 1.09 ms. However, as stated previously, the main focus of this manuscript is to demonstrate the potential of our BP ink for the development of different, novel inkjet-printed novel devices, rather than specific device performance.

Figure R6. Time response of Gr/Si and BP/Gr/Si.

We have included the following additional text in the revised manuscript:

In addition, BP/Gr/Si exhibits a fast response (~ 0.55 ms),

We have also included Fig. R6 as Supplementary Fig. 16(b, c) and the discussion in Supplementary Note 8:

Supplementary Fig. 16(b) presents the time response of Gr/Si and BP/Gr/Si at 1550 nm. Gr/Si shows no response, whereas BP/Gr/Si exhibits a stable and reproducible response. The response time of the cycle shown in Fig. 16(c) is ~ 0.55 ms, and the recovery time is ~ 1.09 ms. The response/recovery time are faster than those previously reported BP photodetectors, for instance ~ 5 ms at 633 nm.

9. e) measurements as a function of the air exposure time should also be provided to confirm stability (see points above).

We thank the reviewer for the comment here. We have measured the photocurrent changes and the time response of BP/Gr/Si at 1550 nm after 7 days exposure to ambient conditions; Fig. R7. As shown, we find negligible changes in both the photocurrent changes and the time response. This demonstrates the stability of our device.

We have included the following additional text in the revised manuscript:

and a long-term (>7 days) operation stability under ambient conditions.

We have also included Fig. R7 as Supplementary Fig. 17 and the discussion in Supplementary Note 8:

We further assess the operation stability of BP/Gr/Si. Supplementary Fig. 17 presents the photocurrent changes and the time response of BP/Gr/Si at 1550 nm after 7 days' exposure to ambient conditions. As shown, we find negligible changes in both the photocurrent changes and the time response. This demonstrates the high operation stability of the device.

Figure R7. BP/Gr/Si photodetection performance after 7 days exposed in open air: (a) photocurrent change; (b) time response.

Reviewer #2

The manuscript by Hu et al. reports an inkjet printing process for liquid exfoliated BP. To my knowledge, inkjet printing of BP has not yet been demonstrated. On the downside, there are some major weaknesses in the results and analysis that need to be addressed before any recommendation can be made:

1. The Raman intensity ratio of $I(A_{1g})/I(A_{2g})$ have seen to be polarization dependent in several reports other than the paper by Favron et al.; examples are: i) Anisotropic Electron-Photon and Electron-Phonon Interactions in Black Phosphorus, ii) Rediscovering black phosphorus as an anisotropic layered material for optoelectronics and electronics, and iii) High-Quality Black Phosphorus Atomic Layers by Liquid-Phase Exfoliation. In these reports, although A_{1g} and A_{2g} follow a similar trend, but the ratio is indeed polarization dependent, and in some angles can easily exceed 1. Favron et al. have shown that A_{1g} peak diminishes earlier than A_{2g} peak during the oxidation which is mainly due to the nature of degradation in the BP and its effect on the crystal vibrational modes. Although they have also reported a very rough insensitivity for the $I(A_{1g})/I(A_{2g})$ to the polarization, but in the light of the other reports, polarization insensitivity needs to be independently verified by the authors before making any conclusion on the oxidation.

We thank the reviewer for this comment. We also acknowledge that whilst Favron *et al.* suggest that the ratio is polarisation independent,⁶ the other works cited by the reviewers (“Anisotropic Electron-Photon and Electron-Phonon Interactions in Black Phosphorus” - Ref. 9 in this response, “Rediscovering black phosphorus as an anisotropic layered material for optoelectronics and electronics” - Ref. 10 in this response, and “High-Quality Black Phosphorus Atomic Layers by Liquid-Phase Exfoliation” - Ref. 11 in this response) do provide evidence to the contrary.

We have conducted our own polarised Raman measurement to shed further light on this; Fig. R8. The sample for this polarised Raman measurement is prepared by dropcasting the BP ink onto a Si/SiO₂ substrate, subsequently dried under nitrogen. Raman characterisation of the BP sample is taken at one single point, using an excitation wavelength of 514 nm with a power <0.1 mW and a duration of 10 s for each polarised angle. The peak intensities here are normalised to $I(A_{2g}^2)$. As shown, the spectra do not show any observable variations under the varied polarisation angles, suggesting that Raman spectra are independent on polarisation. The relationship between the peaks, A_{1g}^1 and A_{2g}^2 , has been used as an indication for BP oxidation.^{6,12} The acquired peak intensity ratio, $I(A_{1g}^1)/I(A_{2g}^2)$ remains constant (0.48 ± 0.04) under the varied polarisation angles (Fig. R8(b)), demonstrating that $I(A_{1g}^1)/I(A_{2g}^2)$ is also polarisation independent. We suggest the other reason accounting for this discrepancy with other studies is that we are studying dropcast and printed BP flakes and as such, there is no alignment in the orientation of the deposited BP flakes. During the measurement, within the area of the laser spot ($\sim 1 \mu\text{m}$) there are many flakes distributed in random orientation to one another. As we have demonstrated, this seems to nullify any polarisation dependency that is otherwise observed in the cited works by the reviewer here.

Figure R8. (a) Polarisation-resolved Raman spectra of dropcast dried BP ink, intensity normalised to $I(A_{2g}^2)$; (b) the associated peak intensity ratio, $I(A_{1g}^1)/I(A_{2g}^2)$. The green region corresponds to low oxidation, and the yellow region high oxidation.

We have made the following changes in the revised manuscript:

BP exhibits highly anisotropic electron-phonon interactions, making the Raman peaks polarisation-, wavelength- and thickness-dependent. → Previous studies show that BP exhibits highly anisotropic electron-phonon interactions, making the Raman peaks polarisation-, wavelength- and thickness-dependent. However, we show that this polarisation behaviour can be nullified when the studied BP flakes (dropcast randomly distributed BP flakes in this case) are not aligned in orientation (Supplementary Fig. 3(a)).

Favron *et al.* showed the intensity ratio of A_g^1 and A_g^2 , $I(A_g^1)/I(A_g^2)$, under 532 nm excitation, is polarisation insensitive and can indicate the oxidation levels of BP flakes, with a range 0.2-0.6 for minimal oxidation. → The intensity ratio of A_g^1 and A_g^2 , $I(A_g^1)/I(A_g^2)$ which we demonstrate to be polarisation insensitive (Supplementary Fig. 3(b)), has been used as an indication for the oxidation levels of exfoliated BP flakes, with a range 0.2-0.6 for minimal oxidation.

We have also included Fig. R8 as Supplementary Fig. 3 and the discussion in Supplementary Note 2:

We conduct polarised Raman measurement to check whether Raman spectrum and $I(A_g^1)/I(A_g^2)$ are polarisation independent or not for our solution processed BP flakes and printed BP samples. The sample for this polarised Raman measurement is prepared by dropcasting the BP ink onto a Si/SiO₂, subsequently dried under nitrogen. Polarised Raman characterisation of the BP sample is taken at one single point, using an excitation wavelength of 514 nm with a power <0.1 mW and a duration of 10 s for each polarised angle. Supplementary Fig. 3(a) presents the polarisation-resolved Raman spectra. The peak intensities here are normalised to $I(A_g^2)$. As shown, the spectra do not show any observable variations under the varied polarisation angles, suggesting that Raman spectra are independent on polarisation. The acquired peak intensity ratio, $I(A_g^1)/I(A_g^2)$ remains constant (0.48 ± 0.04) under the varied polarisation angles (Supplementary Fig. 3(b)), demonstrating that $I(A_g^1)/I(A_g^2)$ is also polarisation independent. We suggest the reason accounting for this polarisation independence here is that we are studying dropcast and printed BP flakes and as such, there is no alignment in the orientation of the deposited BP flakes. During the measurement, within the area of the laser spot ($\sim 1 \mu\text{m}$) there are many flakes distributed in random orientation to one another. As we have demonstrated, this seems to nullify any polarisation dependence that is otherwise observed in literature.

2. The AFM images shown in Fig. 2g are extremely low mag and the height scale bar is chosen so small that the data falls mainly outside of the bounds and show saturation (black or white). These figures are not informative regarding the uniformity of the printing. The comparison to NMP is also not very informative. It is well known and trivial that high boiling temperature solvents cannot be directly used for printing.

We acknowledge that indeed the AFM images in Fig. 2(g) seems of low magnification, and that the BP flakes are not large enough to be observed clearly on this scale unless zoomed in. However, in showing the images, we aim to discuss the flake distribution over the scale of an ink droplet, to provide complementary information to the optical microscope images in Fig. 2(f) and not to study the morphologies of individual BP flakes. As shown in Supplementary Fig. 2(c), the average thickness of the produced BP flakes is 3.37 nm, and 61.5% of the flakes have a thickness ≤ 3 nm. Therefore, we have intentionally set a very low height scale (up to 3 nm) such that these thin flakes are observable in the image and the distribution of all flakes can be properly assessed regardless of the flake thicknesses. By processing the images in this style, we think they sufficiently demonstrate the flake distribution uniformity, and thus confirm the absence of a coffee ring effect when printing with our ink.

It is true that high boiling point solvents such as NMP should not be directly used for printing, indeed, we aim to stress throughout the manuscript the importance of formulating our ink with low boiling point solvents. However, NMP has been widely used in inkjet printing and other solution-based processing of 2d-materials in the literature, for instance inkjet printing of graphene^{1,2,13} and TMDs.^{2,13} Therefore, we think it is necessary to provide comparative evidence between NMP based dispersions and our BP ink. Correspondingly, in various places in the manuscript we have underlined and demonstrated the unsuitability of NMP for inkjet printing in terms of droplet jetting, ink drying, substrate wetting and the coffee ring effect.

3. In general, the use of alcohols as suitable solvents for inkjet printing is also demonstrated, as pointed out by the authors. However, stability of the BP dispersions has neither been reported before, nor characterized by the authors. If BP flakes re-aggregate in IPA, the ink would not be viable. I suggest the authors check the optical absorption coefficients of BP in IPA after a long sitting time.

We thank the reviewer for this comment. Alcohols are indeed widely used as the solvents in printing, and we have demonstrated the suitability of alcohols (*e.g.* IPA/2-butanol) for our BP ink formulation. However, we acknowledge that pure IPA is not the best solvent for BP production, as observed in our results. The carrier solvent combination of IPA and 2-butanol, though, affords the production of a highly-concentrated BP ink with stability over a period of weeks. We accept however that the long-term stability of the ink against reaggregation was not explicitly demonstrated in our manuscript.

We have developed a homemade stability measurement system to address this comment. We employ a 632 nm laser beam through the ink (diluted to 5 vol.% to avoid absorption saturation), and collect the laser intensity transmitted through the diluted ink over one week. The acquired light intensity that is absorbed by the BP flakes, *i.e.* the base laser intensity subtracted by the laser intensity transmitted through IPA/2-butanol, is plotted as normalised absorption in Fig. R9. The absorbed intensity shows only a 1% drop over 180 hours, indicating <1% BP flakes sediment. This demonstrates the high stability of the ink against sedimentation over a timeframe that would prove viable for large-scale ink production and printing.

Figure R9. Ink absorption against time, the ink is diluted to 5 vol.%.

We have made the following changes in the revised manuscript:

The flakes can be redispersed through a brief (10 minutes) sonication in this binary solvent-based ink and is stable against sedimentation for weeks. → **The optical absorption of the formulated ink shows only 1% drop across one week (Supplementary Fig. 8(a)), demonstrating the high stability of the ink against sedimentation over a timeframe that would prove viable for large-scale ink production and printing.**

We have also included Fig. R9 as Supplementary Fig. 8(a) and the discussion in Supplementary Note 3:

As demonstrated, the ink carrier (IPA/2-butanol) affords the production of a highly-concentrated BP ink. However, before conducting printing processes, it is necessary to assess the stability of ink against sedimentation. We develop a homemade stability measurement system to address this. We employ a 632 nm laser beam through the ink (diluted to 5 vol.% to avoid absorption saturation), and collect the laser intensity transmitted through the diluted ink over one week with 5 mins interval. The laser intensity transmitted through the ink carrier, IPA/2-butanol, is also collected as the base laser intensity. The acquired light intensity absorbed by the BP flakes, *i.e.* the difference between the base laser intensity and the laser intensity transmitted through the diluted ink, is plotted as normalised absorption in Supplementary Fig. 8(a). The absorbed intensity shows only a 1% drop over 180 hours, indicating <1% BP flakes sediment. This demonstrates the high stability of the ink against sedimentation over a timeframe that would prove viable for large-scale ink production and printing.

Here are some minor comments/questions about the manuscript:

4. What was the real time difference between the three cases of drying of droplets while measuring the contact angle?

We thank the reviewer for this comment. The real time difference for the contact angle measurements are ~12 hour for NMP dispersion, ~30 s for the IPA_{S.E.} dispersion and ~30 s for the ink also. The volume of the droplets in all three cases is ~2 μL, and the temperature is 20°C.

We have updated the manuscript with the following changes:

faster drying time of the ink and the BP-IPA_{S.E.} (<100 s), when compared to the NMP dispersion (12 hours) → **faster drying time of the ink and the BP-IPA_{S.E.} (~30 s), when compared to the NMP dispersion (~12 hours)**

5. What was the effect of the process on blockage of nozzle? If clogging happened, how soon it happened and what is the potential solution?

As demonstrated above, the IPA/2-butanol mixture affords a BP ink stable against sedimentation over a timescale of weeks. Therefore, during printing, we did not usually observe blockages of the cartridge nozzles due to ink aggregation. An accepted guideline for stable inkjet printing is that the average particle size should be <1/50th of the nozzle diameter (22 μm).¹ The

AFM measurements of our flakes indicate that they are ~ 80 nm in lateral size (Supplementary Fig. 2(b)), significantly smaller than this threshold. The ink therefore allows stable, long, large-scale printing processes. We have uploaded a supplementary video taken during a long printing session (over 6 hours) on a printing scale of $100 \text{ mm} \times 63 \text{ mm}$. To prevent build-up of BP flakes on the nozzles (which could ultimately lead to clogging) across different printing sessions, we conduct 2-3 cleaning cycles of the nozzles *via* purging the nozzles with the IPA/2-butanol mixture before and after each printing session, using the printer's built-in cleaning cycles.

To clarify the printing stability of the ink, we have included this discussion in Supplementary Note 3:

As demonstrated above, the IPA/2-butanol mixture affords a BP ink stable against sedimentation over a timescale of weeks. An accepted guideline for stable inkjet printing is that the average particle size should be $< 1/50$ th of the nozzle diameter ($22 \mu\text{m}$). The AFM measurements of our flakes indicate that they are ~ 80 nm in lateral size (Supplementary Fig. 2(b)), significantly smaller than this threshold. The combination of the ink stability against sedimentation and the small size nature of the flakes therefore allows stable, long, large-scale printing processes. We have uploaded a supplementary video taken during a long printing session (over 6 hours) on a printing scale of $100 \text{ mm} \times 63 \text{ mm}$. To prevent build-up of BP flakes on the nozzles (which could ultimately lead to clogging) across different printing sessions, we conduct 2-3 cleaning cycles of the nozzles *via* purging the nozzles with the IPA/2-butanol mixture before and after each printing session, using the printer's built-in cleaning cycles.

6. How did the process of adding secondary alcohol, 2-butanol, influence the solution concentration? How did the authors estimate the final concentration of the solution after the last step of solvent addition?

We transfer the ink in three steps: (1) sedimentation of BP flakes in NMP *via* high speed centrifugation ($275,000g$); (2) redispersion of the BP flakes in IPA, of which the volume is 10 vol.% of the removed NMP. This dispersion is termed as $\text{IPA}_{S.E.}$ in the manuscript; (3) further add 2-butanol at 10 vol.%. Therefore, the concentration of the $\text{IPA}_{S.E.}$ is 10 times that of the NMP dispersion ($\sim 0.54 \text{ gL}^{-1}$), estimated as $\sim 5.4 \text{ gL}^{-1}$. Adding the 2-butanol dilutes the $\text{IPA}_{S.E.}$ dispersion concentration by 10%, therefore the concentration of the final ink is estimated as $\sim 5 \text{ gL}^{-1}$. We employ UV-Vis optical extinction spectra to verify the concentration of the final ink; Fig. R10. The ink is diluted by 100 times for the measurement. Since the extinction at 660 nm is 0.133, we verify the ink concentration as $\sim 5 \text{ gL}^{-1}$ using the extinction coefficient, $267 \text{ Lg}^{-1}\text{m}^{-1}$ at 660 nm.

Figure R10. Optical extinction (log-log scale) of the ink, the ink is diluted to 1 vol.%.

We have made the following changes in the revised manuscript:

with a concentration $\sim 5 \text{ gL}^{-1} \rightarrow$ with a concentration $\sim 5 \text{ gL}^{-1}$ (verified *via* optical extinction; Supplementary Fig. 7(a))

We have also included Fig. R10 as Supplementary Fig. 7(a) and the discussion in Supplementary Note 3:

After ink formulation, we also employ UV-Vis optical extinction spectrum to verify the concentration of the final ink; Supplementary Fig. 7(a). The ink is diluted to 1 vol.% for the measurement to avoid absorbance saturation. Since the extinction value at 660 nm is 0.133, we verify the ink concentration as $\sim 5 \text{ gL}^{-1}$ using the extinction coefficient, $267 \text{ Lg}^{-1}\text{m}^{-1}$ at 660 nm.

7. How did authors find the optimized amount of secondary alcohol, 2-butanol? How the amount of 2-butanol affects the surface tension gradient of the jetted droplets and coffee ring residues?

We have investigated our ink formulation with varied 2-butanol volume percentages. Here we attach optical micrographs for dried droplets formulated with 20 vol.% 2-butanol, and for 0 vol.% and 10 vol.% ($\text{IPA}_{S.E.}$ and the ink dried droplets quoted from Fig. 2(f) in the manuscript, respectively); Fig. R11(a). The droplets are all $\sim 10 \text{ pL}$ and inkjet-printed onto untreated

Si/SiO₂ and dried at 60°C. As we can observe, 0 vol.% forms a noticeable coffee ring effect, while both 10 vol.% and 20 vol.% do not. The lack of coffee ring suggests that a surface tension gradient is generated to induce Marangoni flow within the droplets in both these cases. We further study the time-dependant contact angle of the three formulations; Fig. R11(b). Contact angle for 0 vol.% and 10 vol.% are quoted from Fig. 2(c) in the manuscript for the IPA_{S,E}. and the ink, respectively. The absence of large variations for 10 vol.% and 20 vol.% confirms the lack of coffee ring effect in these two cases. However, the droplet diameter increases as the volume percentage of 2-butanol increases, with ~75 μm for 10 vol.% and ~85 μm for 20 vol.%. As shown in Fig. R11(b), the contact angle of 20 vol.% decreases faster than that of 10 vol.% during the drying process, suggesting that 20 vol.% spreads faster than 10 vol.%. This may account for the larger dried diameter of 20 vol.%. An increase in drop diameter is undesirable as it means a decrease in printing resolution. Based on the above considerations on the coffee-ring effect and the printing resolution, we chose 10 vol.% for the ink formulation.

Figure R11. (a) Optical micrograph of dried droplets on untreated Si/SiO₂ formulated with 0 vol.%, 10 vol.% and 20 vol.% 2-butanol; (b) Change in contact angle during droplet (~2 μL) drying process on untreated Si/SiO₂ at room temperature.

We have made the following changes in the revised manuscript:

This secondary alcohol is included to produce a composition variation across the droplet during the drying process to induce an inward Marangoni flow. → **The 10 vol.% secondary alcohol is included not only to produce a composition variation across the droplet during the drying process to induce an inward Marangoni flow, but also to preserve a high printing resolution; see Supplementary Fig. 6 and the associated discussion in Supplementary Note 3.**

We have also included Fig. R11 as Supplementary Fig. 6 and the discussion in Supplementary Note 3:

We have investigated our ink formulation with varied 2-butanol volume percentages. Here we attach optical micrographs for dried droplets formulated with 0 vol.%, 10 vol.%, and 20 vol.% 2-butanol; Supplementary Fig. 6(a). 0 vol.% is IPA_{S,E}, and 10 vol.% is the ink in the manuscript; see Fig. 2(f). The droplets are all ~10 pL and inkjet-printed onto untreated Si/SiO₂ and dried at 60°C. As we can observe, 0 vol.% forms a noticeable coffee ring effect, while both 10 vol.% and 20 vol.% do not. The lack of coffee ring suggests that a surface tension gradient is generated to induce Marangoni flow within the droplets in both these cases. We further study the time-dependant contact angle of the three formulations; Supplementary Fig. 6(b). Contact angle for 0 vol.% and 10 vol.% are quoted from Fig. 2(c) in the manuscript. The absence of large variations for 10 vol.% and 20 vol.% confirms the lack of coffee ring effect in these two cases. However, the droplet diameter increases as the volume percentage of 2-butanol increases, with ~75 μm for 10 vol.% and ~85 μm for 20 vol.%. As shown in Supplementary Fig. 6(b), the contact angle of 20 vol.% decreases faster than that of 10 vol.% during the drying process, suggesting that 20 vol.% spreads faster and in a larger area than 10 vol.%. This explains the larger dried diameter of 20 vol.%. Evidently, an increase in drop diameter is undesirable as it means a decrease in printing resolution. Based on the above considerations on the coffee-ring effect and the printing resolution, we chose 10 vol.% for the ink formulation.

8. How different was the result on hydrophilic and hydrophobic substrates? (Si or SiO₂)

We have shown highly uniform, reproducible and controllable printing of the ink on Si/SiO₂, glass and PET without the need for surface treatments in the submission. To clearly demonstrate how hydrophilicity or hydrophobicity defines the printing morphology, we print the ink onto untreated Si/SiO₂, Si/SiO₂ treated with O₂ plasma (duration 60 s, plasma power 100 W), and Si/SiO₂ etched with hydrofluoric acid (HF, duration 5 mins); Fig. R12.

Figure R12. Contact angle and dark field optical micrographs of printed lines on (a, d) untreated Si/SiO₂, (b, e) Si/SiO₂ treated with O₂ plasma, and (c, h) Si/SiO₂ etched with hydrofluoric acid. (d) quoted from Fig. 3(d) in the manuscript.

Treating Si/SiO₂ with O₂ plasma will increase the hydrophilicity of the substrate.¹⁴ Therefore, as shown in Fig. R12(a, b), the ink forms a smaller contact angle on O₂ plasma Si/SiO₂ than untreated Si/SiO₂. This indicates that the ink will spread more on O₂ plasma Si/SiO₂. As a result, indeed, the ink forms a line with a larger width on O₂ plasma Si/SiO₂; Fig. R12(d, e). We also observe that printing on O₂ plasma Si/SiO₂ forms a non-uniform flake distribution where the flakes are mostly confined within a narrow region inside the broad line. Therefore, O₂ plasma leads to a decrease in the printing uniformity and the printing resolution, as has been demonstrated by previous study.¹ On the other hand, etching Si/SiO₂ with HF to completely remove the oxide layer will expose the Si, which is hydrophobic in nature.¹⁵ We etch the Si/SiO₂ with HF for 5 mins, aiming to obtain a highly hydrophobic Si substrate. However, we find HF Si/SiO₂ still forms a smaller contact angle, and as a result a much broader line than the untreated Si/SiO₂. Also, we can observe that the flakes are not uniformly distributed. The printing results here show that the HF Si/SiO₂ substrate is actually hydrophilic. We suggest the reason accounting for the transition from hydrophobic to hydrophilic may be that the exposed Si surface has been oxidised once contacted with ambient oxygen. And also since the surface has been removed with any impurities and organic contaminations, this HF treated Si/SiO₂ turns out to be more hydrophilic than the original untreated Si/SiO₂.

9. In the second page of the manuscript, in the last paragraph before results section, it is stated that, “In addition, the low boiling point of alcohols leads to their fast evaporation and rapid ink drying (<10 s at <60 C) without requiring high temperature curing. This combination of few printing repetitions and rapid drying ensures that the time for BP oxidation during the printing process significantly minimized.” The last sentence is somehow ambiguous. In a way it seems to convey that, as the time of oxidation is minimized, BP oxidizes faster which is not the case. The faster drying process, provides low possibilities of BP oxidation.

We thank the reviewer for this comment, and we apologise for the confusion. As they have correctly interpreted, we did mean to say that the faster drying process leads to a reduced window of time in which BP oxidation is possible.

We have rephrased the last sentence, and thus suggest the following changes in the revised manuscript:

This combination of few printing repetitions and rapid drying ensures that the time for BP oxidation during the printing process significantly minimized. → **A fast printing process due to the reduced printing repetitions and the rapid ink drying leads to a reduced window of time available for the oxidation of BP under ambient conditions during printing.**

Reviewer #3

This paper presents details of the production of black phosphorous (bP) nano-sheets by liquid-phase exfoliation, and the formulation into inks suitable for ink-jet printing. The authors show that the printed black phosphorous can be stabilised against oxidation by encapsulation, over time periods of up to 30 days. These inks have then been used to produce demonstrator devices in optical applications, namely mode-locked laser and photodetector. While liquid-phase exfoliation of bP has been shown before, the optimisation of the dispersions to inkjet printing applications is novel, and will be of interest to researchers working in the field of 2D materials and printed electronics. Importantly, it demonstrates that liquid-phase exfoliation can be used to produce such devices, which is important for potential industrial applications and scale-up.

The manuscript is well written and clearly explains the experimental work and the implications of the results generated. However, the following points should be addressed to further strengthen the paper, and widen the potential audience.

1. The results of the optical absorption spectroscopy are shown as the extinction and scattering spectra, and a peak is claimed in the extinction spectrum. This peak is not clearly visible, however I suspect that it will be more visible in the pure absorption spectrum, and as such I would like to see this presented. This would potentially also give an indication of the band gap of the exfoliated material. Given samples have been measured in an integrating sphere spectrometer, this data should have already been gathered.

We thank the reviewer for this suggestion. The ~ 465 nm peak is wide and is not clearly visible, though it is recognised as a 'peak' in literature.¹² Here we attach the absorbance spectra measured with an integrating sphere; Fig. R13. As can be seen, the absorbance spectra do not reveal any clear peak. We therefore are not able to acquire more information about the bandgap from absorption spectra.

Figure R13. Optical absorbance (log-log scale) of the NMP, CHP and IPA BP dispersions.

To clarify this, we have included Fig. R13 as Supplementary Fig. 1(b) and the discussion in Supplementary Note 1: Supplementary Fig. 1(b) presents the optical absorbance spectra of the produced NMP, CHP and IPA based BP dispersions. Subtracting the extinction data shown in Fig. 1(b) with the absorbance data allows us to obtain the dispersion scattering information presented in Fig. 1(c). Similar to the extinction spectra in Fig. 1(b), the absorbance spectra here also show a broad peak at ~ 465 nm peak, although as this is not clearly resolved, it is not possible to further interpret the absorbance spectra to determine the electronic properties of the material.

2. The use of the scattering spectra to characterise the flake size is a useful approach. It would be useful to apply this technique to the dispersions after solvent transfer to the IPA/butanol mixture to probe for any flake aggregation before printing. Also, do the results here agree with the scattering exponent to length fit presented by Hanlon et al (Nat. Comm. 2015)? And if not, is there an explanation for the discrepancy?

We thank the reviewer for the suggestion and agree that investigation on the scattering spectra of the ink can be useful to assess any possible flake aggregation associated with the ink formulation processes. As requested, we attach the normalised scattering spectra for the ink and the NMP BP dispersion before the ink formulation; Fig. R14. The ink is diluted to 1 vol.% to avoid absorption saturation. Here 1 vol.% is used to keep the diluted ink concentration consistent with that used in scattering

analysis of the BP NMP dispersion, which is 10 vol.%. The normalised scattering of the ink shows a difference compared to that of the NMP dispersion. However, both the normalised scattering spectra can be fitted with a scattering exponent of ~ 1.9 . This suggests that the BP flakes in the ink do not have large variations in flakes sizes, and therefore that the ink formulation procedures do not cause aggregation of the BP flakes.

Figure R14. Optical scattering with associated fitting (log-log scale) of the NMP BP dispersion and the ink diluted to 1 vol.%, with scattering normalised to the 465 nm peak.

Hanlon *et al.* (Nat. Comm. 2015)¹² did investigate the relationship between scattering and selected BP flake sizes. The BP flakes were selected via control of the centrifugation speed. Quoting the supplementary information of this cited paper, “The scattering exponent was obtained by fitting the scattering spectra from 700-800 nm plotted on a log-log plot to a linear relation to obtain the scattering exponent - n' as slope of the fit...The nanosheet length can thus be determined from the scattering spectra by L (μm) = $0.42 / n'$.” We use n' here is to distinguish it from the scattering exponent symbol, n , we use in the submission. Applying this equation to our own data, the length for our BP flakes in NMP and in the ink is estimated as ~ 220 nm, inconsistent with our estimation (80-145 nm) and the average lateral dimension (80.46 nm) we measured *via* AFM.

There are several reasons that could account for this discrepancy: (1) We and the authors may have used different fitting equations. The authors in Hanlon *et al.* did not actually explain the formula used to fit the scattering data to the wavelength within the 700-800 nm wavelength range. We fit our data following the equation $\alpha \propto \lambda^{-n}$, where α is the measured scattering, n is the scattering exponent, and λ is the wavelength where the scattering is linearly dependent on the wavelength on the log-log scale.^{16,17} Therefore, n (used by us) and n' (used by Hanlon *et al.*) may have different definition. (2) The authors Hanlon *et al.* investigated a wavelength region at 700-800 nm, whereas we investigate the wavelength region where the scattering is linearly dependent on the wavelength on the log-log scale, *i.e.* ~ 465 -1000 nm. This defines how the scattering data is fitted. Therefore, we may conclude with different scattering exponents assuming we are based on the same fitting method. (3) The authors Hanlon *et al.* investigated the scattering of BP flakes after size selection, whereas we investigate the scattering of BP flakes without size selection. Therefore, assuming we both used the same fitting method, the authors studied the dependence of scattering on selected flake sizes, whereas we studied the dependence of the scattering as a whole on the average flake size.

To clarify this, we have included the following additional text in the revised manuscript:

The optical scattering of the ink (Supplementary Fig. 7(b)) does not exhibit a significant difference from the BP NMP dispersion, indicating that the ink formulation processes do not induce any aggregation of the BP flakes.

We have also included Fig. R14 as Supplementary Fig. 7(b) and the discussion in Supplementary Note 3:

After ink formulation, we again investigate the scattering spectrum of the produced ink to assess whether there are any possible flake aggregations associated with the solvent transfer process. Supplementary Fig. 7(b) presents the scattering spectra and associated scattering fitting for the ink and the BP NMP dispersion. The ink is diluted to 1 vol.% to avoid absorption saturation, and 1 vol.% is used here also to keep the diluted ink concentration consistent with that of the BP NMP dispersion, which is 10 vol.% diluted for measurement. The normalised scattering of the ink shows a difference compared to that of the NMP dispersion. However, both the spectra can be fitted with a scattering exponent of ~ 1.9 . This suggests that the BP flakes in the ink do not have large variations in flakes sizes, and therefore that the ink formulation procedures do not cause aggregation of the BP flakes.

3. The discussion of the raman maps in fig. 2e and 4d state that "there are no localised regions with [intensity ratio] > 0.6." However both the maps and corresponding histograms show that there are locations with a ratio over 0.6. The authors should clarify this apparent discrepancy. (break it as 3(a) and put it after the first reply pls) Also, although the cited paper (Favron et al.) shows that ratios lower than 0.2 are indicative of higher levels of oxidation, there is no evidence shown there that values above 0.6 also indicate oxidation. The present authors claim that values >0.6 indicate oxidations, and so a clearer explanation of this is needed. Perhaps XPS measurements would confirm degree of oxidation that can then be correlated to the raman measurements.

We thank the reviewer for this comment and apologise for the confusion. We acknowledge that there are locations with ratio >0.6 in the Raman maps in Fig. 2(e) and Fig. 4(c). However, these >0.6 (and the <0.2) are isolated data points that are randomly distributed within the investigated Raman mapping areas. This is why we used the phrase "no localised regions" in our original manuscript, to mean that the oxidation of BP is occurring randomly over the entire investigated area, rather than in clusters.

To clarify this, we have made the following change in the revised manuscript:

localised regions with $I(A_g^1)/I(A_g^2) > 0.6$. → **localised clusters where $I(A_g^1)/I(A_g^2) > 0.6$.**

The optical scattering of the ink (Supplementary Fig. 7(b)) does not exhibit a significant difference from the BP NMP dispersion, indicating that the ink formulation processes do not induce any aggregation of the BP flakes.

Regarding the ratio value threshold for oxidation, we quote Fig. 6(e) (See Fig. R15(a) here) and page 5 of "Photooxidation and quantum confinement effects in exfoliated black phosphorus" - Ref. 6 in this response: "The statistic reveals that values of $A_g^1/A_g^2 > 0.2$ are characteristic of low oxidation levels (yellow region)"; and "it shows that pristine samples have ratios in the range 0.4–0.6". Based on this cited work, 0.2-0.6 is characteristic of low oxidation, and therefore we consider >0.6 (and <0.2) as an indication for a high oxidation. However, we note that there are other studies (for instance, "Liquid exfoliation of solvent-stabilized few-layer black phosphorus for applications beyond electronics" - Ref. 12 in this response, which consider >0.6 for low oxidation and <0.6 for high oxidation.

We have independently conducted our own experiments *via* Raman mapping to investigate BP oxidation of freshly cleaved BP crystal through mechanical exfoliation (as presented in the submission). We plot the histogram of the intensity ratio, $I(A_g^1)/I(A_g^2)$; Fig. R15(b-d). For the freshly cleaved bulk BP sample, in which case high oxidation is very unlikely, the majority of the acquired ratio values are observed within 0.2-0.6, with only 3.1% >0.6. For the NMP dispersion and printed BP that with higher possibilities of oxidation due to the exfoliation, ink formulation and printing processes, the percentage for >0.6 increases up to 4.18% and 10.03%, respectively. This independently suggests that >0.6 indicates a high oxidation, and 0.2-0.6 a low oxidation. We thus use 0.2-0.6 as the threshold for characterization of BP oxidation.

We acknowledge that XPS is definitely a reliable method to assess the oxidation of individual BP flakes. However, we think the Raman mapping is also a reliable and well-suited method for studying solution processed, printed as well as encapsulated BP flakes than XPS for direct comparison.

To clarify this, we have included this discussion in Supplementary Note 2:

Here we note that though a ratio value of 0.2-0.6 is used as the threshold for BP low oxidation by Favron *et al.*, there are other studies (for instance, Hanlon *et al.*) which consider >0.6 for low oxidation and <0.6 for high oxidation. We have independently analysed the $I(A_g^1)/I(A_g^2)$ Raman mapping histograms for freshly cleaved bulk BP (Supplementary Fig. 13(a)), BP NMP dispersion (Fig. 1(f)), and printed BP (Fig. 4(d)). These BP samples represent three different oxidation status. For the freshly cleaved bulk BP sample, in which case high oxidation is very unlikely, the majority of ratio values are observed within 0.2-0.6, with only 3.1% >0.6. For the NMP dispersion and printed BP that with higher possibilities of oxidation due to the exfoliation, ink formulation and printing processes, the percentage for >0.6 increases up to 4.18% and 10.03%, respectively. This, in conjunction with the above references, suggests that >0.6 indicates a high oxidation, and 0.2-0.6 a low oxidation. We thus use 0.2-0.6 for minimal oxidation.

Figure R15. (a) Quoting Fig. 6(e) of “Photooxidation and quantum confinement effects in exfoliated black phosphorus” - Ref. 6 in this response; Histogram of the intensity ratio, $I(A_g^1)/I(A_g^2)$ for (b) freshly cleaved bulk BP, (c) dried dropcast BP NMP dispersion, and (d) BP ink after printed out.

4. Did the authors attempt to exfoliate directly in the IPA/Butanol mixture that was used for inkjet printing? Given the resulting dispersions are stated as being stable against sedimentation for weeks, I would be interested to know if this was attempted, and if so, how it compared to the other solvents. Avoiding the solvent exchange step would make the process more attractive to commercial scale-up.

We did indeed attempt to exfoliate BP directly in the IPA/2-butanol mixture. Here we attach the optical extinction spectrum of the resulting dispersion (diluted to 10 vol.% to avoid saturation); Fig.R16. The spectrum allows us to estimate the concentration of the dispersion as $\sim 0.35 \text{ gL}^{-1}$. The NMP, CHP and IPA dispersions have a concentration $\sim 0.54 \text{ gL}^{-1}$, $\sim 0.32 \text{ gL}^{-1}$ and $\sim 0.13 \text{ gL}^{-1}$, respectively. This shows that the IPA/2-butanol performs better than CHP and IPA, but not as well as NMP. As demonstrated in our original manuscript, the ink formulation from NMP based dispersion, after solvent exchange, allows an ink concentration as high as $\sim 5 \text{ gL}^{-1}$, nearly 15 times the concentration of the IPA/2-butanol dispersion. To avoid BP oxidation during printing, a highly-concentrated ink is required to significantly reduce the number of printing repetitions. Therefore, at present, we view that solvent exchange as an unavoidable compromise if we are to achieve a high concentration ink, even though with the as-prepared IPA/2-butanol BP dispersion which is suitable for inkjet printing.

However, since this is not particularly relevant to the scope of this manuscript, we decide not to include this discussion here into the manuscript or the Supplementary Information.

Figure R16. Optical absorbance (log-log scale) of the IPA/2-butanol dispersion (10 vol% diluted), the inset showing the diluted dispersion.

5. Can the authors clarify the technique used for the contact angle measurements? The methods state a pendant drop technique, but the discussion on p. 5, paragraph 3 talks about deposition of material at the droplet edge, and contact line sticking. This suggest that sessile drop approach has been used.

We thank the reviewer for bringing this to our attention. We did state in the manuscript that we used the pendant droplet technique to measure the surface tension of the ink. However, we failed to mention the contact angle measurement details in the manuscript or in the Supplementary Information. We did indeed use the sessile drop technique to measure the contact angle.

We have now clarified this in the revised manuscript Methods section:

For contact angle measurement, a $\sim 2 \mu\text{L}$ droplet is dropcast and measured *via* sessile drop technique. The measurement is conducted at room temperature ($\sim 20^\circ\text{C}$).

6. Fig. 4a shows a noticeable deviation from the linear trend for the first and second printing passes. Do the authors have an explanation for this?

We thank the reviewer for this comment. The optical absorption for 1 and 2 printing repetitions is very low as can be seen from the printed patterns in the inset figure. We have done repeated absorbance measurements for the initial 2 printing repetitions (including of other printed 2d materials such as graphene), and they all exhibit the same absorbance behaviour, with variation $< 5\%$. We believe that at such low extinction, the acquired data is subjected to ambient noise which might have been overcompensated by the spectrometer.

7. The study on the stability of printed material, with and without encapsulation is clearly important, and highly relevant for potential users of this material and techniques. What would be equally important, and would improve the relevance of the paper would be equivalent measurement of the inks themselves. This would allow an estimation of a shelf-life for inks, which may be produced separate from the printing location.

We thank the reviewer for this comment. Regarding the stability of the ink against sedimentation, we have developed a homemade stability measurement system to address this comment. We employ a 632 nm laser beam through the ink (diluted to 5 vol.% to avoid absorption saturation), and collect the laser intensity transmitted through the diluted ink over one week. The acquired light intensity that is absorbed by the BP flakes, *i.e.* the base laser intensity subtracted by the laser intensity transmitted through IPA/2-butanol, is plotted as normalised absorption in Fig. R17(a). The absorbed intensity shows only a 1% drop over 180 hours, indicating $< 1\%$ BP flakes sediment. This demonstrates the high stability of the ink against sedimentation over a timeframe that would prove viable for large-scale ink production and printing.

Regarding the ink stability against oxidation, we have conducted Raman $I(A_g^1)/I(A_g^2)$ ratio mapping of a dried dropcast ink sample using a formulation prepared two months previously and stored under nitrogen in the interim. Here we attach the acquired intensity ratio histogram; Fig. R17(b). It shows that the proportion outside 0.2-0.6 has increased to 21.69%, from the contribution of data points with intensity ratio > 0.6 . Therefore, whilst this is not an excessive increase given the timescale, it is clear that it is best to use freshly prepared BP ink for device fabrications.

We suggest that the oxidation of the ink may arise from the trace water in the ink solvents. The solvents used were purchased in

Figure R17. (a) Ink absorption against time, the ink is diluted to 5 vol.%; (b) $I(A_g^1)/I(A_g^2)$ histogram for dropcast dried BP ink, the ink has been kept under nitrogen for 2 months; (c) Optical absorbance (log-log scale) of the IPA/2-butanol ink carrier.

anhydrous composition, with the aim of minimising water content. However, the measured optical absorbance (Fig. R17(c)) of the ink carrier, IPA/2-butanol, shows notable water signals in the near infrared wavelength region, *i.e.* the peaks at $\sim 0.91 \mu\text{m}$, $\sim 1.01 \mu\text{m}$, $\sim 1.19 \mu\text{m}$ and $\sim 1.39 \mu\text{m}$. This suggests that there was trace water introduced into the ink carrier during the handling and formulation processes. We believe that this trace water contributes to the degradation to BP in the stored ink. If it is indeed the case, we suggest that it is possible to have long shelf life with our BP ink as long as the ink formulation and subsequent storage takes place in a controlled atmosphere.

We have made the following changes in the revised manuscript:

The flakes can be redispersed through a brief (10 minutes) sonication in this binary solvent-based ink and is stable against sedimentation for weeks. → The optical absorption of the formulated ink shows only 1% drop across one week (Supplementary Fig. 8(a)), demonstrating the high stability of the ink against sedimentation over a timeframe that would prove viable for large-scale ink production and printing.

We have also included Fig. R17 as Supplementary Fig. 8 and the discussion in Supplementary Note 3:

As demonstrated, the ink carrier (IPA/2-butanol) affords the production of a highly-concentrated BP ink. However, before conducting printing processes, it is necessary to assess the stability of ink against sedimentation. We develop a homemade stability measurement system to address this. We employ a 632 nm laser beam through the ink (diluted to 5 vol.% to avoid absorption saturation), and collect the laser intensity transmitted through the diluted ink over one week with 5 mins interval. The laser intensity transmitted through the ink carrier, IPA/2-butanol, is also collected as the base laser intensity. The acquired light intensity absorbed by the BP flakes, *i.e.* the difference between the base laser intensity and the laser intensity transmitted through the diluted ink, is plotted as normalised absorption in Supplementary Fig. 8(a). The absorbed intensity shows only a 1% drop over 180 hours, indicating $<1\%$ BP flakes sediment. This demonstrates the high stability of the ink against sedimentation over a timeframe that would prove viable for large-scale ink production and printing.

We now assess the stability of ink itself against oxidation. We have conducted Raman $I(A_g^1)/I(A_g^2)$ ratio mapping of a dried dropcast ink sample using a formulation prepared two months previously and stored under nitrogen in the interim. Here we attach the acquired intensity ratio histogram; Supplementary Fig. 8(b). It shows that the proportion outside 0.2-0.6 has increased to 21.69%. Therefore, whilst this is not an excessive increase given the timescale, it is clear that it is best to use freshly prepared BP ink for device fabrications.

We propose that this oxidation of the ink may arise from the trace water in the ink carrier. The ink carrier solvents, *i.e.* IPA and 2-butanol, used were purchased in anhydrous composition, with the aim of minimising water content. However, the measured optical absorbance (Supplementary Fig. 8(c)) of the ink carrier also shows notable water signal peaks at $\sim 0.91 \mu\text{m}$, $\sim 1.01 \mu\text{m}$, $\sim 1.19 \mu\text{m}$ and $\sim 1.39 \mu\text{m}$. This suggests that there was trace water introduced into the ink carrier during the handling and formulation processes. We believe that this trace water contributes to the degradation to BP in the stored ink. If it is indeed the case, we argue that it is possible to have long shelf life with our BP ink as long as the ink formulation and subsequent storage takes place in a controlled atmosphere.

8. The performance over time of the saturable absorber has been demonstrated, and so it would be interesting to see similar results from the photodetector. For those not familiar with the specific applications demonstrated, it would be helpful to include, if possible, a comparison with current technologies, and/or a discussion of the advantages that inkjet black phosphorous based devices offer. This would help to widen the potential audience of the paper.

We thank the reviewer for the comment here. We have measured the photocurrent changes and the time response of BP/Gr/Si at 1550 nm after 7 days exposure to ambient conditions; Fig. R18. As shown, we find negligible changes in both the photocurrent changes and the time response. This demonstrates the stability of our device.

Figure R18. BP/Gr/Si photodetection performance after 7 days exposed in open air: (a) photocurrent change; (b) time response.

We have included the following additional text in the revised manuscript:
and a long-term (>7 days) operation stability.

We have also included Fig. R18 as Supplementary Fig. 17 and the discussion in Supplementary Note 8:
We further assess the operation stability of BP/Gr/Si. Supplementary Fig. 17 presents the photocurrent changes and the time response of BP/Gr/Si at 1550 nm after 7 days exposure to ambient conditions. As shown, we find negligible changes in both the photocurrent changes and the time response. This demonstrates a high operation stability of the device.

State-of-the-art applications of BP in optoelectronics and photonics:

(1) Saturable absorbers (SA): The nonlinear absorption properties of BP make it a promising material candidate for the development of saturable absorbers (SA) for ultrafast lasers, as discussed in the manuscript. There are already reports of such devices fabricated both through solution-processing based techniques and otherwise; Table 1. To the best of our knowledge, we are the first to report printable BP SAs. The SAs show excellent device operation under an intense irradiation for over 30 days. The stable operation period is much longer than those reported.

(2) Photodetectors: With a direct bandgap of 0.3-2.0 eV spanning the visible to infrared wavelength region, BP holds huge potential for visible to infrared photodetection applications. Such applications have been reported already; Table 2. However, these devices are fabricated with mechanically exfoliated BP flakes. This material production technique suffers from extremely low yield and high uncontrollability. Therefore, this device fabrication technique is highly limited by the material production yield, and it requires high device fabrication complexity and cost. To the best of our knowledge, this is the first report of printable BP photodetectors. The produced visible to near-infrared photodetector exhibits high responsivities (up to 164 mA/W), fast response (up to ~ 0.55 ms), and a long-term (>7 days, *i.e.* 168 hours) operation stability. Also, this printable technology enables the benefits of high yield, low cost, reduced fabrication complexity, as well as potentially the thin-form factor, flexibility, and stretchability of the fabricated photodetectors.

In addition, printable BP devices also hold potential in:

(3) Electronics: There are reports on BP based transistors.^{41,42} These devices exhibit high electron mobility (up to 1,000 $\text{cm}^2\text{V}^{-1}\text{s}^{-1}$) and drain current modulation (up to 10^5). However, these devices are also fabricated using mechanically exfoliated BP flakes. Similarly, this device fabrication technique is limited by low fabrication yield, high cost, and high fabrication complexity. There are no reports on BP based printable transistors. However, we envision our BP ink has a high potential to

Table 1. Mode-locked fibre lasers using BP SAs. ME - mechanical exfoliation; LPE - liquid phase exfoliation; α_d - modulation depth; λ , operating wavelength; τ , pulse duration.

Fabrication method	α_d (%)	Laser type	Laser properties		Demonstrated stability (hours)	Reference
			λ (nm)	τ		
Inkjet printing	8.7	Er: Fibre	1562	605 fs	>714	Our work
ME	8.1	Er: Fibre	1571	946 fs	28	18
ME	7.5	Er: Fibre	1561	272 fs	-	19
ME	0.6-4.6	Tm: Fibre	1910	739 fs	-	20
ME	-	Er: Fibre	1559	786 fs	-	21
ME	9.8	Er: ZBLAN	2783	42 ps	-	22
ME	8	Yb: Fibre	1086	7.54 ps	-	23
ME	6.9	Er: Fibre	1561	2.66 ps	-	24
ME	-	Er: Fibre	1559	805 fs	-	25
LPE	21	Er: Fibre	1560	670 fs	-	26
LPE	0.8	Er: Fibre	1561	1.44 ps	-	27
LPE	41.2	Ho/Pr : ZBLAN	2867	8.6 ps	-	28
LPE	6.91	Er: Fibre	1532-1570	940 fs	10	29
LPE	4.1	Er: Fibre	1558	2.18 ps	-	30
LPE	50-90	Tm/Ho: Fibre	1880-1940	1.58 ps	2	31
LPE	19	Er: Fibre	1568	117.6 ns	-	32
LPE	10.1	Er: Fibre	1569	280 fs	24	33

Table 2. Photodetector based on BP. ME - mechanical exfoliation; LPE - liquid phase exfoliation; IR - infrared; FET - field effect transistor

Fabrication	Structure	Properties			Demonstrated stability (hours)	Reference
		Spectral range	Responsivity (mA/W)	Response time		
Inkjet printing	Schottky junction	visible - near IR	164	0.55 ms	>168	Our work
ME	FET	visible - near IR	20	-	-	34
ME	FET	visible - near IR	4.8	1 ms	-	35
ME	FET	mid IR	82	-	-	36
ME	FET	terahertz	0.15 (V/W)	-	-	37
ME	Waveguide	near IR	657	-	-	7
ME	p-n junction	visible - near IR	1.5	40 μ s	-	38
ME	Heterojunction	visible	418	-	-	39
ME	Heterojunction	visible - near IR	22.3	15 μ s	-	40

develop printable electronics, which indeed has been successfully demonstrated with other 2d materials.^{1,8}

(4) Energy storage: Solution processed BP flakes show potential as electrode material for batteries and supercapacitors, as reported in preceding studies.^{43,44} There are no reports on BP based printable batteries or supercapacitors. We also envision that, with the BP ink, it is possible to achieve printable energy storage devices, with successful demonstrations with other functional materials.^{45,46}

To highlight the huge potential of inkjet printing of BP in applications, we have made the following changes in the revised manuscript:

We note this operation time is over 72 times longer than any previously reported. → We compare our work with previous reports (Supplementary Table 2), and show that the stability of our device is >25 times longer.

We have also included the following additional text in the revised manuscript:

Supplementary Table 3 presents the comparison between our work and the current BP photodetection technology based on different device structures, for instance waveguide that enables a record high responsivity. It is thus highly potential to integrate

our BP ink into such structures to allow large-scale fabrication of devices with further enhanced performance.

We have also included the Table 1 and 2 and the associated discussions in the Supplementary Information:

Table 1 presents the reported results in literature of BP SAs fabricated both through solution-processing based techniques and otherwise as a comparison to our work. The SAs show excellent device operation under intense irradiation for over 714 hours, at least 25 times longer than those previously reported.

Application of BP in photodetectors have been reported already; Table 2. However, these devices are fabricated with mechanically exfoliated BP flakes. This material production technique suffers from extremely low yield and high uncontrollability. Therefore, this device fabrication technique is highly limited by the material production yield, and it requires high device fabrication complexity and cost. To the best of our knowledge, this is the first report of printable BP photodetectors. The produced visible to near-infrared photodetector exhibits high responsivities, fast response (up to ~ 0.55 ms), and a long-term (>7 days) operation stability. Also, this printable technology enables the benefits of high yield, low cost, reduced fabrication complexity, as well as potentially the thin-form factor, flexibility, and stretchability of the fabricated photodetectors.

Besides the SAs and photodetector we have demonstrated already *via* inkjet printing of BP, we envision that our BP ink formulation also holds huge potential in other applications, such as printable electronics and printable energy storage. Though there are no reports of such BP based printable devices yet, we have seen many successful demonstrations of proof-of-concepts based on printing of other functional material systems. For instance, inkjet printed graphene transistors developed by Torrisi *et al.*, all inkjet printed 2d material based read-only memories developed by McManus *et al.*, and all inkjet printed carbon nanotubes based flexible supercapacitors developed by Choi *et al.*. The BP ink can be effortlessly transferred to the fabrication of such devices.

References

1. Torrisi, F. *et al.* Inkjet-printed graphene electronics. *ACS Nano* **6**, 2992–3006 (2012).
2. Finn, D. J. *et al.* Inkjet deposition of liquid-exfoliated graphene and MoS₂ nanosheets for printed device applications. *J. Mater. Chem. C* **2**, 925–932 (2014).
3. Curcio, J. A. & Petty, C. C. The near infrared absorption spectrum of liquid water. *J. Opt. Soc. Am.* **41**, 302 (1951).
4. Wozniak, B. & Dera, J. Light absorption by water molecules and inorganic substances dissolved in sea water. In *Light Absorpt. Sea Water*, 11–81 (Springer, New York, NY, 2007).
5. Castellanos-Gomez, A. *et al.* Isolation and characterization of few-layer black phosphorus. *2D Mater.* **1**, 025001 (2014).
6. Favron, A. *et al.* Photooxidation and quantum confinement effects in exfoliated black phosphorus. *Nat. Mater.* **14**, 826–832 (2015).
7. Youngblood, N., Chen, C., Koester, S. J. & Li, M. Waveguide-integrated black phosphorus photodetector with high responsivity and low dark current. *Nat. Photonics* **9**, 247 (2015).
8. McManus, D. *et al.* Water-based and biocompatible 2D crystal inks for all-inkjet-printed heterostructures. *Nat. Nanotechnol.* (2017).
9. Ling, X. *et al.* Anisotropic electron-photon and electron-phonon interactions in black phosphorus. *Nano Lett.* **16**, 2260–2267 (2016).
10. Xia, F., Wang, H. & Jia, Y. Rediscovering black phosphorus as an anisotropic layered material for optoelectronics and electronics. *Nat. Commun.* **5** (2014).
11. Yasaei, P. *et al.* High-quality black phosphorus atomic layers by liquid-phase exfoliation. *Adv. Mater.* **27**, 1887–1892 (2015).
12. Hanlon, D. *et al.* Liquid exfoliation of solvent-stabilized few-layer black phosphorus for applications beyond electronics. *Nat. Commun.* **6**, 8563 (2015).
13. Withers, F. *et al.* Heterostructures produced from nanosheet-based inks. *Nano Lett.* **14**, 3987–3992 (2014).
14. Alam, A. U., Howlader, M. M. R. & Deen, M. J. The effects of oxygen plasma and humidity on surface roughness, water contact angle and hardness of silicon, silicon dioxide and glass. *J. Micromechanics Microengineering* **24**, 035010 (2014).
15. Bal, J., Kundu, S. & Hazra, S. Hydrophobic to hydrophilic transition of HF-treated Si surface during Langmuir-Blodgett film deposition. *Chem. Phys. Lett.* **500**, 90–95 (2010).
16. O'Neill, A., Khan, U. & Coleman, J. N. Preparation of high concentration dispersions of exfoliated MoS₂ with increased flake size. *Chem. Mater.* **24**, 2414–2421 (2012).
17. Backes, C. *et al.* Edge and confinement effects allow in situ measurement of size and thickness of liquid-exfoliated nanosheets. *Nat. Commun.* **5**, 4576 (2014).
18. Chen, Y. *et al.* Mechanically exfoliated black phosphorus as a new saturable absorber for both Q-switching and Mode-locking laser operation. *Opt. Express* **23**, 12823 (2015).
19. Sotor, J., Sobon, G., Macherzynski, W., Paletko, P. & Abramski, K. M. Black phosphorus saturable absorber for ultrashort pulse generation. *Appl. Phys. Lett.* **107**, 051108 (2015).
20. Sotor, J. *et al.* Ultrafast thulium-doped fiber laser mode locked with black phosphorus. *Opt. Lett.* **40**, 3885 (2015).
21. Li, D. *et al.* Polarization and thickness dependent absorption properties of black phosphorus: New saturable absorber for ultrafast pulse generation. *Sci. Rep.* **5**, 15899 (2015).
22. Qin, Z. *et al.* Mid-infrared mode-locked pulse generation with multilayer black phosphorus as saturable absorber. *Opt. Lett.* **41**, 56 (2016).
23. Hisyam, M. B., Rusdi, M. F. M., Latiff, A. A. & Harun, S. W. Generation of Mode-Locked Ytterbium doped fiber ring laser using few-layer black phosphorus as a saturable absorber. *IEEE J. Sel. Top. Quantum Electron.* **23**, 39–43 (2017).
24. Ismail, E. I., Kadir, N. A., Latiff, A. A., Ahmad, H. & Harun, S. W. Black phosphorus crystal as a saturable absorber for both a Q-switched and mode-locked erbium-doped fiber laser. *RSC Adv.* **6**, 72692–72697 (2016).
25. Lee, D., Park, K., Debnath, P. C., Kim, I. & Song, Y.-W. Thermal damage suppression of a black phosphorus saturable absorber for high-power operation of pulsed fiber lasers. *Nanotechnology* **27**, 365203 (2016).

26. Song, Y. *et al.* Vector soliton fiber laser passively mode locked by few layer black phosphorus-based optical saturable absorber. *Opt. Express* **24**, 25933 (2016).
27. Mao, D. *et al.* Stable high-power saturable absorber based on polymer-black-phosphorus films. *Opt. Commun.* (2016).
28. Li, J. *et al.* Black phosphorus: a two-dimension saturable absorption material for mid-infrared Q-switched and mode-locked fiber lasers. *Sci. Rep.* **6**, 30361 (2016).
29. Luo, Z.-C. *et al.* Microfiber-based few-layer black phosphorus saturable absorber for ultra-fast fiber laser. *Opt. Express* **23**, 20030–20039 (2015).
30. Park, K. *et al.* Black phosphorus saturable absorber for ultrafast mode-locked pulse laser via evanescent field interaction. *Ann. Phys.* **527**, 770–776 (2015).
31. Yu, H., Zheng, X., Yin, K., Cheng, X. & Jiang, T. Thulium/holmium-doped fiber laser passively mode locked by black phosphorus nanoplatelets-based saturable absorber. *Appl. Opt.* **54**, 10290 (2015).
32. Chen, Y. *et al.* Optically driven black phosphorus as a saturable absorber for mode-locked laser pulse generation. *Opt. Eng.* **55**, 081317 (2016).
33. Chen, Y., Chen, S., Liu, J., Gao, Y. & Zhang, W. Sub-300 femtosecond soliton tunable fiber laser with all-anomalous dispersion passively mode locked by black phosphorus. *Opt. Express* **24**, 13316 (2016).
34. Engel, M., Steiner, M. & Avouris, P. Black phosphorus photodetector for multispectral, high-resolution imaging. *Nano Lett.* **14**, 6414–6417 (2014).
35. Buscema, M. *et al.* Fast and broadband photoresponse of few-layer black phosphorus field-effect transistors. *Nano Lett.* **14**, 3347–3352 (2014).
36. Guo, Q. *et al.* Black phosphorus mid-infrared photodetectors with high gain. *Nano Lett.* **16**, 4648–4655 (2016).
37. Viti, L. *et al.* Black phosphorus terahertz photodetectors. *Adv. Mater.* **27**, 5567–5572 (2015).
38. Yuan, H. *et al.* Polarization-sensitive broadband photodetector using a black phosphorus vertical p-n junction. *Nat. Nanotechnol.* **10**, 707–713 (2015).
39. Deng, Y. *et al.* Black phosphorus-monolayer MoS₂ van der Waals heterojunction p-n diode. *ACS Nano* **8**, 8292–8299 (2014).
40. Ye, L., Li, H., Chen, Z. & Xu, J. Near-infrared photodetector based on MoS₂ /black phosphorus heterojunction. *ACS Photonics* **3**, 692–699 (2016).
41. Li, L. *et al.* Black phosphorus field-effect transistors. *Nat. Nanotechnol.* **9**, 372–377 (2014).
42. Zhu, W. *et al.* Flexible black phosphorus ambipolar transistors, circuits and AM demodulator. *Nano Lett.* **15**, 1883–1890 (2015).
43. Chen, L. *et al.* Scalable clean exfoliation of high-quality few-layer black phosphorus for a flexible lithium ion battery. *Adv. Mater.* **28**, 510–517 (2016).
44. Hao, C. *et al.* Flexible all-solid-state supercapacitors based on liquid-exfoliated black-phosphorus nanoflakes. *Adv. Mater.* **28**, 3194–3201 (2016).
45. Choi, K.-H., Yoo, J., Lee, C. K. & Lee, S.-Y. All-inkjet-printed, solid-state flexible supercapacitors on paper. *Energy Environ. Sci.* **9**, 2812–2821 (2016).
46. Kaempgen, M., Chan, C. K., Ma, J., Cui, Y. & Gruner, G. Printable thin film supercapacitors using single-walled carbon nanotubes. *Nano Lett.* **9**, 1872–1876 (2009).

Reviewers' Comments:

Reviewer #1:

Remarks to the Author:

The authors have revised the work following the recommendations of the referees. The answers provided to my comments are solid and extra data are now included in the SI - therefore I recommend publication of this manuscript.

I have only a minor comment regarding the use of water as solvent - water based phosphorene dispersions have been demonstrated by the Hersam group (Stable aqueous dispersions of optically and electronically active phosphorene, Jooheon Kang, DOI: 10.1073/pnas.1602215113). The problem is not the water, but the use of surfactants as stabilising agent, as also explained in the introduction by the authors. I would advise the authors to rephrase the related sentence (pag 5).

Reviewer #2:

I believe the authors' responses are satisfactory to most of my comments, except the comment on AFM imaging. The manuscript entirely lacks any direct microscopic information on the morphology and quality of the printed flakes. The authors have provided histograms of lateral and thickness distribution of the flakes from AFM studies which have been most likely obtained through extensive imaging. Such a high spatial resolution (in which 50 nm and 60 nm flakes can be told apart) means lots of high resolution AFM images are available to the authors. I suggest the authors to provide data for review and representative images in the paper.

Reviewer #3:

Remarks to the Author:

The authors have revised the manuscript to address the comments I made in my review, and have made sufficient additions and clarifications to the points raised. On a couple of points however, there is additional revisions that I feel are needed. Referring to the number from that initial review:

1. I still do not feel the data presented in the figures, including the absorption spectra in the additional figure S1, support the claim of a peak in the spectrum at 465nm. I had expected that such a peak would be seen in the absorption spectrum, when the scattering component has been removed. The fact that the clear absorption peaks are not seen in the current work, in contrast to that shown by Hanlon et al., seems strange. Although the extinction spectra are similar, the absorption spectra are quite different. I am concerned slightly that this also affects the scattering spectra that have been presented here. I therefore feel this issue needs further attention.

6. If the divergence from linear is due to limitations of the spectrometer, then the authors should include a quantification of the uncertainty in these measurements.

The expanded discussion of the state-of-the-art of black phosphorous makes the manuscript much more accessible to a wider audience.

Point by Point Response to the Reviewers' Comments

Reviewer #1

The authors have revised the work following the recommendations of the referees. The answers provided to my comments are solid and extra data are now included in the SI - therefore I recommend publication of this manuscript.

I have only a minor comment regarding the use of water as solvent - water based phosphorene dispersions have been demonstrated by the Hersam group (Stable aqueous dispersions of optically and electronically active phosphorene, Jooheon Kang, DOI: 10.1073/pnas.1602215113). The problem is not the water, but the use of surfactants as stabilising agent, as also explained in the introduction by the authors. I would advice the authors to rephrase the related sentence (pag 5).

We thank the reviewer for this comment.

To clarify this, we have made the following changes in the revised manuscript:

However, BP tends to degrade in water, therefore this water-based ink formulation is not well-suited for BP. → However, developing stable aqueous BP dispersions requires deoxygenated water assisted by surfactants. This demands additional processing steps to remove the surfactants following printing.

Reviewer #2

I believe the authors' responses are satisfactory to most of my comments, except the comment on AFM imaging. The manuscript entirely lacks any direct microscopic information on the morphology and quality of the printed flakes. The authors have provided histograms of lateral and thickness distribution of the flakes from AFM studies which have been most likely obtained through extensive imaging. Such a high spatial resolution (in which 50 nm and 60 nm flakes can be told apart) means lots of high resolution AFM images are available to the authors. I suggest the authors to provide data for review and representative images in the paper.

We thank the reviewer for this comment. As the reviewer points out, the AFM images in the manuscript (Fig. 2(g)) are acquired by large area scans ($80\ \mu\text{m} \times 80\ \mu\text{m}$) and are thus not suitable for the study of the morphology and quality of the flakes. We reiterate that these are intended as a representation of the flake distributions over an entire deposited droplet, as complementary information to the optical microscope images (Fig. 2(f)), confirming the absence of a coffee ring effect when printing with our ink. To clarify with respect to the reviewers comment, in Supplementary Note 1, the lateral dimension and thickness statistics (Supplementary Fig. 2(d-i)) are based on analysis of a large number of flakes over multiple, smaller, higher resolution scans ($10\ \mu\text{m} \times 10\ \mu\text{m}$). We had not included these scans in the manuscript as we felt that the data drawn from them is conveyed more appropriately by the histograms, whilst the uniformity of flake distribution was better displayed by the $80\ \mu\text{m} \times 80\ \mu\text{m}$ whole-droplet scans.

In response, we include Fig. R1(a), which presents representative high-resolution AFM images for the individual thin BP flakes. These images have been taken by scanning smaller areas within the $10\ \mu\text{m} \times 10\ \mu\text{m}$ sample areas to isolate individual flakes. As observed, all these flakes show clean surfaces with clearly defined edges, further suggesting that the flakes in the ink are not oxidised. The lateral dimension of these BP flakes is varied between 70 nm and 400 nm, while the thickness is typically 4-10 nm; Fig. R1(b). We note that this size is relatively large compared to the statistics presented in Supplementary Fig. 2(d, g). This is because the AFM measurements are taken under ambient conditions and as such, slow, high resolution imaging of individual, smaller, thinner flakes is challenging due to their increased rate of oxidation (we note here that faster scans, of lower, but sufficient, resolution, are used for the aforementioned gathering of size and thickness distributions). Indeed, we observe that in Fig. R1(a) the larger, thicker flakes have sharper edges and more distinct morphologies than the smaller flakes, owing to the size and thickness-dependent rate of degradation over the duration of the scan.

Figure R1. AFM study: (a) Representative images for BP flakes and (b) corresponding height profiles, scale bar - 50 nm.

To clarify this, we have included Fig. R1 in Supplementary Fig. 2 with discussion in the revised Supplementary Note 1: Supplementary Fig. 2(j) presents representative AFM images for individual thin BP flakes. As observed, all these flakes show

clean surfaces with clearly defined edges, indicating that the flakes in the ink are not oxidised. The lateral dimension of these BP flakes is varied between 70 nm and 400 nm, while the thickness is typically 4-10 nm; Supplementary Fig. 2(k). We note that this size is relatively large compared to the statistics presented in Supplementary Fig. 2(d, g). This is because the AFM measurements are taken under ambient conditions and as such, slow, high resolution imaging of individual, smaller, thinner flakes is challenging due to their increased rate of oxidation (we note here that faster scans, of lower, but sufficient, resolution, are used for the aforementioned gathering of size and thickness distributions). Indeed, we observe that in Supplementary Fig. 2(j) the larger, thicker flakes have sharper edges and more distinct morphologies than the smaller flakes, owing to the size and thickness-dependent rate of degradation over the duration of the scan.

Reviewer #3

The authors have revised the manuscript to address the comments I made in my review, and have made sufficient additions and clarifications to the points raised. On a couple of points however, there is additional revisions that I feel are needed. Referring to the number from that initial review:

1. I still do not feel the data presented in the figures, including the absorption spectra in the additional figure S1, support the claim of a peak in the spectrum at 465nm. I had expected that such a peak would be seen in the absorption spectrum, when the scattering component has been removed. The fact that the clear absorption peaks are not seen in the current work, in contrast to that shown by Hanlon *et al.*, seems strange. Although the extinction spectra are similar, the absorption spectra are quite different. I am concerned slightly that this also affects the scattering spectra that have been presented here. I therefore feel this issue needs further attention.

We thank the reviewer for this comment. We quote the optical absorbance spectra (normalised to 340 nm) of “Liquid exfoliation of solvent-stabilized few-layer black phosphorus for applications beyond electronics” (Fig. 3(e) in Ref. 1); Fig. R2(a). The authors Hanlon *et al.* studied the absorbance of the dispersions with size-selected BP flakes, and showed that the absorbance peak was more prominent for larger flakes. In the last response and supplement, we presented the absorbance spectra of the NMP, CHP and IPA based BP dispersions on log-log scale (Fig. R2(b) in this response) since we aimed to keep it consistent with Fig. 1(b, c) in the manuscript. Indeed, the absorbance spectra in these cases failed to show very clear peaks. Here, we re-plot the absorbance spectra (normalised to 340 nm) on linear scale; Fig. R2(c). As observed, the NMP spectrum shows only a small peak and that for the CHP only slightly larger, whereas a very prominent peak can be observed in the IPA spectrum. This may arise from the BP flake size distributions in these three dispersions, where NMP (lateral dimension 80.46 nm, thickness 3.37 nm) < CHP (lateral dimension 108.09 nm, thickness 6.00 nm) < IPA (lateral dimension 140.24 nm, thickness 6.58 nm) (Supplementary Fig. 2). This is consistent with the observation by Hanlon *et al.*.

Figure R2. (a) Optical absorbance spectra normalised to 340 nm, adapted from Fig. 3(e) of “Liquid exfoliation of solvent-stabilized few-layer black phosphorus for applications beyond electronics” - Ref. 1 in this response; Optical absorbance spectra (b) on log-log scale and (c) normalised to 340 nm on linear scale of the NMP, CHP and IPA based dispersions.

To clarify this, we have also included Fig. R2(c) in Supplementary Fig. 1 and made the following changes in the revised Supplementary Note 1:

Similar to the extinction spectra in Fig. 1(b), the absorbance spectra here also show a broad peak at ~ 465 nm peak, although as

this is not clearly resolved, it is not possible to further interpret the absorbance spectra to determine the electronic properties of the material. → This absorbance spectra, however, fails to show very clear peaks at ~ 465 nm on the log-log scale. We therefore normalise it to 340 nm and re-plot it on linear scale; Supplementary Fig. 1(c). As observed, the NMP spectrum shows only a small peak and that for the CHP only slightly larger, whereas a very prominent peak can be observed in the IPA spectrum. Hanlon *et al.* have shown that this absorbance peak is flake-size dependent, and that it is more prominent for larger flakes. This therefore suggests that the BP flake size distributions in our dispersions are $\text{NMP} < \text{CHP} < \text{IPA}$, which can be confirmed by the AFM statistics of flake lateral dimension and thickness (Supplementary Fig. 2(d-i)).

6a. If the divergence from linear is due to limitations of the spectrometer, then the authors should include a quantification of the uncertainty in these measurements.

We thank the reviewer for this comment. As demonstrated in the manuscript, the overall extinction for the sample with 1-10 printing repetitions shows an overall variation of $< 2\%$. The optical extinction for 1 and 2 printing repetitions, however, shows a relatively large divergence from the linear fitting. We have attributed this to the very low optical extinction in these two cases, lying at the limit of sensitivity for the spectrometer. As presented in Fig. R3, the optical extinction is divergent by $\sim 110\%$ from the fitted extinction for 1 printing repetition and $\sim 30\%$ for 2 printing repetitions. For 3 printing repetitions, it drops to $\sim 5\%$ whilst it is only $\sim 1\%$ for 10 printing repetitions. We therefore suggest use of printing processes with ≥ 3 printing repetitions if high reproducibility is to be ensured.

Figure R3. Optical extinction divergence from the fitted extinction for 1-10 printing repetitions.

To clarify this, we have included the following additional text in the revised manuscript:

We note 1-2 printing repetitions show a large divergence from the linear fitting. We attribute this to the very low optical extinction in these two cases and the limitations in the sensitivity of the spectrometer. We therefore suggest use of printing processes with ≥ 3 printing repetitions if high reproducibility is to be ensured (Supplementary Fig. 9(c)).

We have also included Fig. R3 in Supplementary Fig. 9 with discussion in the revised Supplementary Note 4:

We show in Fig. 4(a) that the optical extinction for 1 and 2 printing repetitions has a relatively large divergence from the linear fitting though the overall variation is $< 2\%$ between 1-10 printing repetitions. As presented in Supplementary Fig. 9(c), the optical extinction is divergent by $\sim 110\%$ from the fitted extinction for 1 printing repetition and $\sim 30\%$ for 2 printing repetitions. For 3 printing repetitions, it drops to $\sim 5\%$ whilst it is only $\sim 1\%$ for 10 printing repetitions.

6b. The expanded discussion of the state-of-the-art of black phosphorous makes the manuscript much more accessible to a wider audience.

We thank the reviewer for this comment. This has indeed expanded the accessibility of our manuscript.

References

1. Hanlon, D. *et al.* Liquid exfoliation of solvent-stabilized few-layer black phosphorus for applications beyond electronics. *Nat. Commun.* **6**, 8563 (2015).

Reviewers' Comments:

Reviewer #2:

Remarks to the Author:

The authors addressed my comments and I would like to recommend this manuscript for the publication.

Reviewer #3:

Remarks to the Author:

The authors have clarified the issue of the peak in the absorption spectra of the dispersions adequately.

Regarding the issue of the extinction as function of printing passes, the authors have not quite addressed my point (which I was perhaps not clear in expressing). If the divergence from the (expected) linear trend is due to reaching the low end of the sensitivity of the spectrometer, then the value used here may simply be the noise floor of the spectrometer. This noise level can be quantified and expressed as an uncertainty in the measurement itself. This is different from the reproducibility of the process (which is also important). The measurement of a blank substrate, and/or a substrate 'printed' with the solvent alone would also be helpful.

Dear Reviewer,

In this new “Point by Point Response to Reviewers’ comments”, we have outlined our rebuttal and the changes we have made in the further revised manuscript. For your convenience, in this document, your comments and questions are written in “bold”, our responses in “black” and modifications in “green”. In the further revised Manuscript and Supplementary Information, we have similarly highlighted all the second revision changes in “green”. Please note that in the further revised manuscript, we have also highlighted the changes in “red” and “blue” we made previously in response to your first and second comments.

Point by Point Response to the Reviewers’ Comments

Reviewer #3

The authors have clarified the issue of the peak in the absorption spectra of the dispersions adequately.

Regarding the issue of the extinction as function of printing passes, the authors have not quite addressed my point (which I was perhaps not clear in expressing). If the divergence from the (expected) linear trend is due to reaching the low end of the sensitivity of the spectrometer, then the value used here may simply be the noise floor of the spectrometer. This noise level can be quantified and expressed as an uncertainty in the measurement itself. This is different from the reproducibility of the process (which is also important). The measurement of a blank substrate, and/or a substrate ‘printed’ with the solvent alone would also be helpful.

We thank the reviewer for the valuable suggestion. We now include the extinction spectrum of the empty chamber of the spectrometer after zero-adjustment in Fig. R1(a). The extinction of the empty chamber is 0 in the investigated wavelength region, with typical spikes of up to ± 0.002 . We consider this as the noise floor when the spectrometer chamber is empty. The measured extinctions of the bare glass substrate and the glass substrate with dried printed ink carrier (*i.e.* printed ink solvents without the BP flakes) are identical across the measured wavelength range (*e.g.* 0.121 at 550 nm); Fig. R1(a). We note that the extinction was already considered as the background and subtracted in Fig. 4(a) in the original manuscript.

Figure R1. (a) Optical extinction spectra of the empty chamber of the spectrometer, the bare glass, and the glass with dried printed ink carrier; (b) Optical extinction (at 550 nm) of printed BP, with the blue bars indicating the noise floor of the empty chamber spectrometer.

In Fig. 4(a) of the original manuscript (replotted as Fig. R1(b) here), we investigate the printing consistency by studying the extinction of printed BP with 1-10 printing repetitions. Though it demonstrates an overall variation of $<2\%$, 1 and 2 printing repetitions have divergences of $\sim 110\%$ and $\sim 30\%$ from the fitted extinction values, respectively, as noted in the revised Supplementary Information. In response to the reviewer’s comment, we now include the noise floor when the spectrometer chamber is empty in Fig. R1(b), as indicated by the blue bars. As shown, this measured noise floor is negligible compared to the extinction of printed BP flakes (after background subtraction, as explained above). We therefore consider that the low-end sensitivity of the spectrometer does not affect the divergences in the case for 1 and 2 printing repetitions.

We attribute the divergence in the measured extinction for the 1 and 2 printing repetitions to a modification of the air-glass interface after printing, which causes a change in the interface-induced loss (*e.g.* reflection) during the measurement. Subsequent printing repetitions only increase the thickness and absorbance of printed BP without changing this interface. Thus, for successive printing repetitions, the trend follows a constant increase with each printing repetition.

To clarify this, we have made the following changes in the revised manuscript:

We attribute this to the very low optical extinction in these two cases and the limitations in the sensitivity of the spectrometer.

→ We attribute this increase in the measured extinction of the 1 and 2 printing repetitions to a modification of the air-glass interface after printing. For successive printing repetitions the trend follows a constant increase with each printing repetition.

Reviewers' Comments:

Reviewer #3:

Remarks to the Author:

I am happy that the authors have now addressed the issue around the non-linearity of the absorption data, and presented an assessment of the uncertainty in the measurements.

I am happy to recommend acceptance for publication.